# A spatiotemporal style transfer algorithm for dynamic visual stimulus generation

Antonino Greco [1,2,3] ✉ & Markus Siegel [1,2,3,4] ✉

Understanding how visual information is encoded in biological and artificial systems often requires the generation of appropriate stimuli to test specific hypotheses, but available methods for video generation are scarce. Here we introduce the spatiotemporal style transfer (STST) algorithm, a dynamic visual stimulus generation framework that allows the manipulation and synthesis of video stimuli for vision research. We show how stimuli can be generated that match the low-level spatiotemporal features of their natural counterparts, but lack their high-level semantic features, providing a useful tool to study object recognition. We used these stimuli to probe PredNet, a predictive coding deep network, and found that its next-frame predictions were not disrupted by the omission of high-level information, with human observers also confirming the preservation of low-level features and lack of high-level information in the generated stimuli. We also introduce a procedure for the independent spatiotemporal factorization of dynamic stimuli. Testing such factorized stimuli on humans and deep vision models suggests a spatial bias in how humans and deep vision models encode dynamic visual information. These results showcase potential applications of the STST algorithm as a versatile tool for dynamic stimulus generation in vision science.

Understanding how visual information is encoded in the brain has been a longstanding goal of neuroscience and vision science. Recently, there has been increasing interest to also study hidden representations of computer vision models and to compare biological and artificial vision[1]. A fundamental component of this research is the generation of controlled visual stimuli. Traditionally, stimulus manipulation in vision research has been performed for low-level features of static visual stimuli such as the matching of pixel contrast[2], or manipulating the speed of dynamic visual stimuli[3,4]. Recently, computer graphics approaches have led to a proliferation of methods that can generate parametrically designed images for both machine learning and neuroscience research. Examples include the parametric generation of facial expressions[5] or three-dimensional (3D) visual scenes[6,7].

Although these methods have progressed the field of stimulus generation to a remarkable degree, they often fall short in terms of flexibility and tend to diverge from the natural statistics of our visual environment[8], often resulting in outputs that can appear artificial. In contrast, deep neural network models (DNNs) trained for computer vision tasks[9,10] have revolutionized the approach of image synthesis in a variety of ways, affording researchers novel paradigms with which to investigate visual processing within artificial and biological neural networks[11,12]. One of the earliest examples of a DNN-based stimulus generation method is DeepDream[13,14], an algorithm that accentuates patterns in images in a manner that can be conceived as 'algorithmic pareidolia', using a pretrained deep convolutional neural network (CNN)[15]. This method has been predominantly used in vision research to simulate the visual hallucination patterns commonly found in altered states of consciousness induced by psychedelic drugs[16–18]. An alternative to this method has also been proposed to generate visual stimuli that maximally activate specific visual cortical regions across

[1]Department of Neural Dynamics and Magnetoencephalography, Hertie Institute for Clinical Brain Research, University of Tübingen, Tübingen, Germany. [2]Centre for Integrative Neuroscience, University of Tübingen, Tübingen, Germany. [3]MEG Center, University of Tübingen, Tübingen, Germany. [4]German Center for Mental Health (DZPG), Tübingen, Germany. ✉e-mail: antonino.greco@uni-tuebingen.de; markus.siegel@uni-tuebingen.de

species[12,19,20]. Another influential family of DNN-based methods for image synthesis comes from the seminal work of Gatys and colleagues[21], in which they proposed the 'neural style transfer' (NST) algorithm to extract and recombine the texture and shape of natural images to generate novel stimuli[21–24]. NST has been applied in vision research to compare visual representations of images in humans and machines[25,26] and to understand how spatial texture information is encoded in the visual cortex of the mammalian brain[27–29].

Despite the abundance of methods for generating static visual stimuli, the algorithms proposed for dynamic visual stimuli generation, namely videos, are scarce. Unlike images, videos also have a temporal dimension and, in natural videos, the relationship between spatial and temporal features is often non-trivial[27,30,31]. Here we introduce a method for generating dynamic visual stimuli that can serve different purposes in vision research. The method is based on the NST algorithm but extends its capabilities to videos. We thus refer to it as 'spatiotemporal style transfer' (STST). We capitalized on recent texturization algorithms that extended the NST capabilities to generate dynamic textures, that is, video clips with stationary spatiotemporal patterns such as flames or sea waves[32,33]. The general idea is to synthesize 'model metamers'[34], which are dynamic stimuli whose layer activations within a DNN model are matched to those of natural videos. To create these stimuli, we perform optimization in a two-stream model architecture[33,35,36]. This model is designed to replicate the relative segregation of the neural pathways responsible for processing spatial and temporal features in the brain[37–39].

After elucidating the procedural steps of the algorithm, we demonstrate its use in generating model metamers from natural videos. These metamers retain low-level spatiotemporal features but lack high-level object semantics, and thus offer a tool to study object recognition in biological and artificial visual systems[40–42]. Using state-of-the-art deep vision models, we tested early and late layer activations for differences between natural and metamer stimuli. We then examined predictive coding networks' internal representations during next-frame prediction and assessed human object recognition on metamer stimuli. Finally, we illustrate how our STST algorithm enables spatiotemporal factorization, blending spatial and temporal features from different videos for discrimination tasks in both humans and deep vision models.

## Results

### The STST algorithm

The proposed algorithm is based on a two-stream neural network model. which we outline here. Details of the model description are presented in the Methods. One module (or 'stream') processes spatial features in each frame, and the other module captures temporal features across consecutive frames (Fig. 1). We adopted VGG-19[43] as the spatial module and the multiscale spacetime-oriented energy model (MSOE)[33,44] as the temporal module. The optimization procedure generates 'model metamers' that match the content and/or texture between generated and target videos by minimizing spatial and temporal losses. This yields four distinct loss components that, when weighted and combined, form a flexible overall loss function, enabling diverse combinations for dynamic stimulus generation.

Crucially, we employed several preconditioning techniques to enhance the perceptual stability and prevent artifacts: the addition of a total variation loss[45], the multiscale nature of the optimization processing[14,46], a color transfer postprocessing[47,48] and a frame blending operation. These techniques enable the algorithm to generate dynamic stimuli that maintain a consistent appearance over time, particularly for complex natural videos.

### Model metamers for object recognition

We now describe an example application of STST for investigating object recognition in visual systems. We generated dynamic metamer stimuli that had low-level spatiotemporal features similar to those of their natural counterparts, but that lacked high-level object

information. We collected three high-quality video clips from the YouTube-8M dataset[49] and optimized dynamic stimuli to match both their spatial and temporal textures (Fig. 2). For comparison, we generated dynamic stimuli starting from the same target videos but using another algorithm[50] that was previously proposed to generate dynamic metamers[34]. This alternate algorithm basically consists of randomizing the phase in the spatiotemporal frequency domain[50]. In the following, we will refer to this alternate algorithm as spatiotemporal phase scrambling (STPS). The natural videos, as well as their STST and STPS counterparts, are available in Supplementary Video 1.

By qualitatively inspecting the generated frames in Fig. 2a, it can be noticed how our STST algorithm and STPS produce very dissimilar stimuli, both in terms of appearance and motion dynamics. We computed four basic spatiotemporal features for the original, STST and STPS videos to quantitatively assess both methods, namely pixel intensity and contrast as spatial features, and pixel change and optical flow (magnitude and angle) as temporal features (Fig. 2b). We found that both methods excelled at matching the spatial features, but the temporal features were less preserved. Critically, STST performed substantially better than STPS at matching the optical flow of the natural videos, which is crucial for dynamic stimuli.

We validated these results on a large sample of 100 videos from the Kinetics400 dataset[51], which, in the following, we refer to as the validation dataset. We computed the Pearson correlation coefficient (PCC; Supplementary Fig. 1a) and Euclidean distance (ED; Supplementary Fig. 1b) between the low-level features of the original and STST or STPS videos to quantitatively measure to what extent these features matched. All four spatial and temporal features were significantly better preserved by STST than by STPS (all temporal features: PCC, $P < 0.0001$, Cohen's $d > 0.76$; ED, $P < 0.001$, $d > 0.34$; all spatial features: PCC, $P < 0.015$, $d > 0.35$; ED, $P < 0.0001$, $d > 1.08$). The strongest effect was found for the optical flow angle (PCC, $d = 4.25$; ED, $d = 1.62$). These results show how the STST algorithm can be used to dismantle high-level image information while preserving low-level spatiotemporal statistics.

We performed an ablation study on several components of the STST algorithm to assess their contribution to this performance of the algorithm. Specifically, we repeated the generation of the same three example videos while deleting either the color-matching step or total variation loss. We found that both ablations substantially deteriorated the quality of the generated metamers (Supplementary Fig. 2). By inspecting the generated frames and the time course of the low-level statistics (Supplementary Fig. 2), it can be seen that the color-matching steps have a strong impact on contrast and luminosity, as well as on the full color distribution, whereas the absence of the total variation loss induces excessive high-frequency noise. Thus, color matching and total variation loss substantially contribute to the performance of the algorithm.

### Metamer effects on layer representations of vision models

We next tested the generated stimuli using state-of-the-art deep learning models for image and video classification. We used these models because they exhibit a hierarchical structure for extracting visual features that is similar to the mammalian visual cortex[9,52]. We thus hypothesized that the similarity between natural and STST stimuli in layer activations of these models should be higher in early layers than in late layers (Fig. 2c). To compute the similarity between layer activations, we used the center kernel alignment (CKA) score[53]. Notably, we did not use the same models we deployed for the spatial and temporal streams in the STST model.

We found almost perfect matching of early-layer representations between natural and metamer stimuli across all example videos in the image classification models (Fig. 2d) (ResNet50 (ref. 54) and ConvNeXt-T[55]). Conversely, late layer activations differed substantially between natural and metamer stimuli. This pattern of decreased

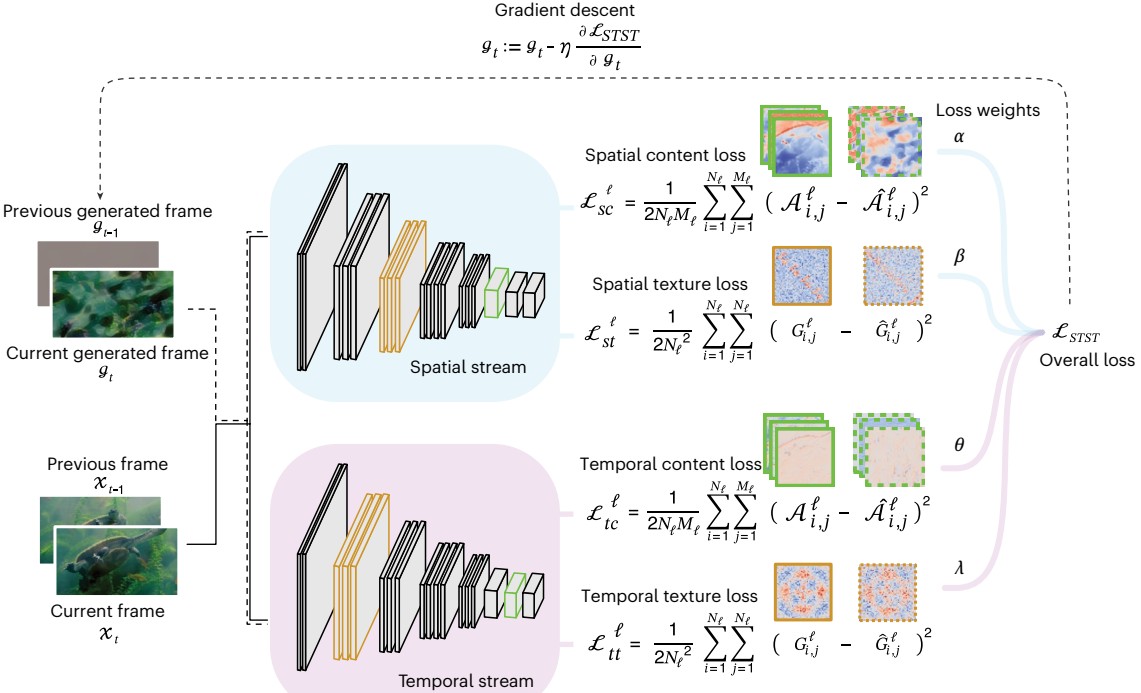

**Fig. 1 | Graphical representation of the STST algorithm.** The current frame as well as the current and previous frames are forward passed to the spatial and temporal stream, respectively, up to a predefined set of layers. Content (green) and texture (orange) loss are defined for both streams as the difference of layer activations (here, green-highlighted layers for the content loss and orange-highlighted layers for the texture loss) and their Gram matrices between the target and generated frames, respectively. These are weighted and summed to form the overall STST loss. The partial derivative of this loss with respect to the current generated frame is used to optimize it by gradient descent. Natural images adapted from Google LLC under a Creative Commons license CC BY 4.0.

similarity was consistent across all videos. Although these results were in line with our predictions, the employed models were applied on a frame-by-frame basis as they were conceived for image classification. Thus, we extended our analyses to video classification models using ResNet18-3D[56] and ResNet18-MC3 (ref. 57). We used the same procedure, but this time using snippets of videos in a sliding-window approach. Again, we observed early-layer activations being very similar across all frames and example videos, whereas late layers showed consistently decreased similarity. Crucially, we found the same pattern as in image classification models, although the dynamic range of the effects was smaller (Fig. 2e).

We validated these findings on the validation dataset (Supplementary Fig. 3a,b). We found that the CKA score was significantly higher in early compared to late layers (for image models, all $P < 0.0001$, all $d > 3.14$; for video models, all $P < 0.0001$, all $d > 1.34$). We computed the CKA score between hidden layer activations of natural and STPS stimuli, finding that for STPS also, early layers had a significantly higher score than late layer representations (for image models, all $P < 0.0001$, all $d > 4.64$; for video models, all $P < 0.0001$, all $d > 2.43$). We also compared the CKA score of layer activations between STST and STPS. For all models and layers, the CKA score was significantly higher in STST than in STPS (all $P < 0.0001$, all $d > 0.94$). Thus, the better preservation of low-level features in STST as compared to STPS may lead to a higher similarity between natural and STST as compared to STPS stimuli, even at later model stages. Together, these findings confirmed that STST allows us to generate dynamic stimuli with high similarity to natural videos in terms of low-level spatiotemporal statistics, but without preserving high-level features that are critical for object recognition in biological and artificial vision systems.

**Evaluating semantic understanding in PredNet with metamers**
We used the metamer stimuli generated with STST to investigate the representational capabilities of DNNs inspired by predictive coding theories[58,59]. We focused on PredNet[60], a deep convolutional recurrent neural network (Fig. 3a) that was trained with self-supervised learning to perform next-frame prediction[61]. We opted for this model because it has interesting properties that are aligned with a range of phenomena observed in the visual cortex[62], probably due to its architectural structure, which involves recurrent and feedback connections[60] that are absent in traditional feedforward-only CNN models[42,63]. Recently, it has been debated whether PredNet is able to extract robust high-level representations of its dynamic inputs or it merely acts as a flow filter predicting low-level optical flow[58,64].

To investigate this, we compared the next-frame predictions of PredNet for both original and STST-generated metamer stimuli (Fig. 3a). We reasoned that if PredNet were able to exploit high-level visual representations, next-frame predictions should be better for natural as compared to metamer stimuli. Figure 3b shows frame predictions by the PredNet model for the three example videos that were also used in the previous experiments. Qualitatively, predictions had a high fidelity compared to their ground truth for both natural and metamer stimuli. We computed the structural similarity index measure (SSIM)[65] to quantify how well PredNet performed next-frame prediction (Fig. 3c).

We found that for the three example videos, SSIM was slightly higher for STST as compared to natural stimuli across frames and on average. Again, to statistically assess these results, we repeated the analysis on the larger validation dataset and conducted the same analysis for stimuli generated with the STPS algorithm (Fig. 3d). We found that SSIM was indeed significantly higher for STST predictions than for natural videos ($P < 0.0001$, $d = 0.29$). Furthermore, for STPS stimuli, SSIM was even higher than for the original ($P < 0.0001$, $d = 1.25$) and STST ($P < 0.0001$, $d = 0.92$) videos. Thus, in contrast to what would be expected if PredNet exploited high-level content, the PredNet predictions were better for STPS stimuli and even better for STST stimuli as compared to natural videos.

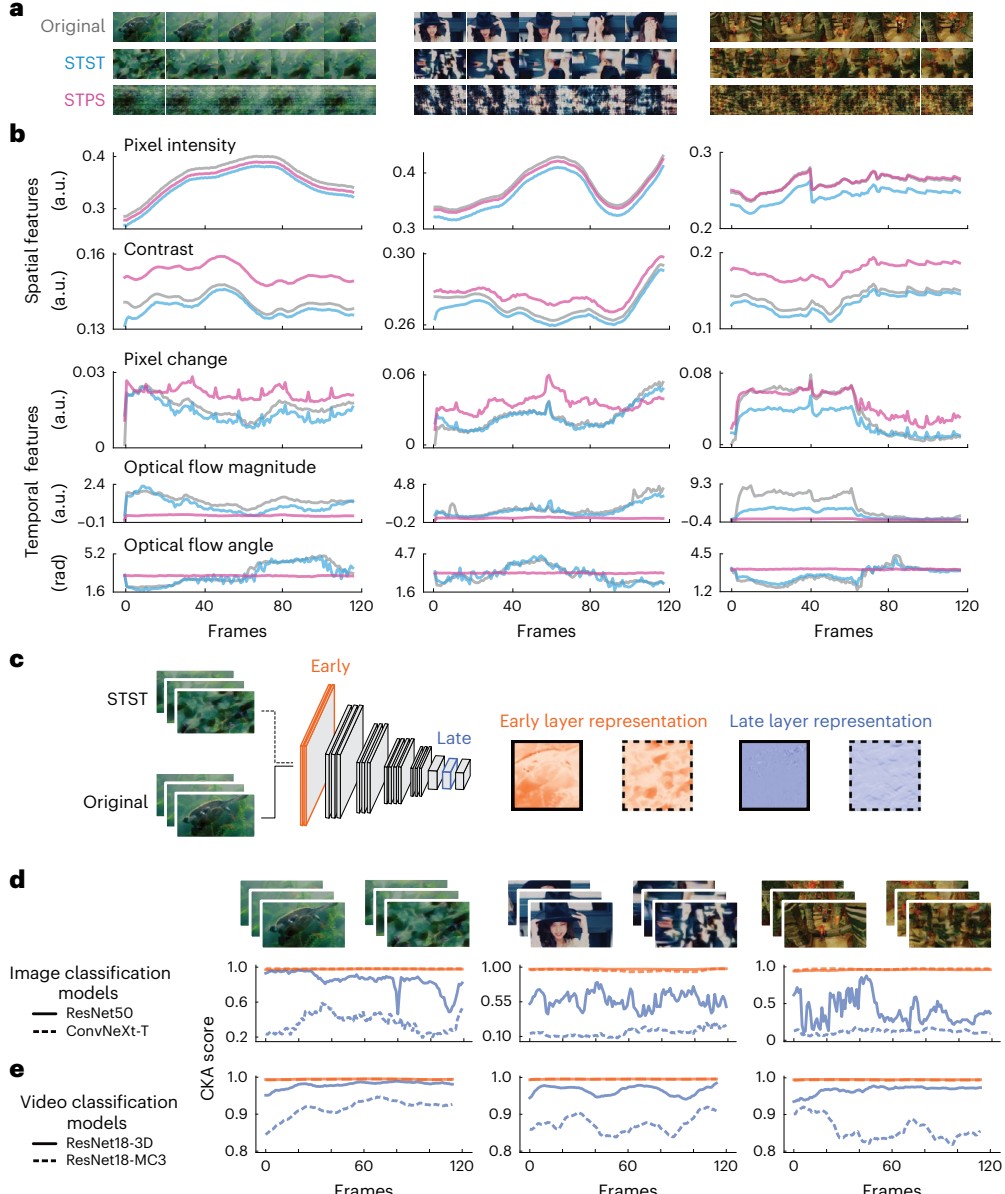

**Fig. 2 | Low-level features in metamers and their effects on layer representations of deep vision models. a**, Example frames from the original target videos as well as from STST- and STPS-generated videos. **b**, Time courses of four spatiotemporal features in original (gray), STST (azure) and STPS (pink) stimuli. Features include pixel intensity and contrast as spatial low-level features and pixel change and optical flow (magnitude and angle) as temporal low-level features. The x axis for all plots is the frame number. On the y axis, a.u. stands for arbitrary units while rad stands for radiants. **c**, We used STST stimuli and their natural counterparts as inputs to deep vision models and extracted early (orange) and late (blue) layer activations. **d**, Time courses of the similarity metric (CKA) between natural and generated stimuli for image classification models in early (orange) and late (blue) layer activations. **e**, Time courses of the similarity metric (CKA) for video classification models in early (orange) and late (blue) layer activations. Natural images in **a**, **c** and **d** adapted from Google LLC under a Creative Commons license CC BY 4.0.

We reasoned that this effect could reflect the extent to which PredNet predictions are merely a copy of the previous frame, which could be more successful if consecutive frames were more alike in the metamer videos than in the original videos. To investigate this, we computed the average SSIM between all consecutive frames in all videos (Fig. 3e). This value was indeed significantly higher for STPS ($P < 0.0001$, $d = 1.62$) and STST ($P < 0.0001$, $d = 0.35$) videos than for natural ones. Importantly, the SSIM between consecutive frames for STST was significantly lower than for STPS ($P < 0.0001$, $d = 1.39$), and thus closer to the natural videos, showing that STST stimuli are better suited to address the research questions at hand. To test whether the increased SSIM for metamer predictions was influenced by this stimulus statistic, we computed a dynamic version of SSIM conditional on the previous frame (cSSIM)[64].

cSSIM scales SSIM by the extent to which next-frame predictions diverge from merely copying the previous frame (Fig. 3f and Methods). Indeed, cSSIM was significantly decreased for STST videos as compared to the original videos ($P = 0.0002$, $d = 0.22$), and was even lower for STPS compared to both original ($P < 0.0001$, $d = 0.51$) and STST ($P < 0.0001$, $d = 0.28$) videos. Together, these results show that PredNet predictions were not impaired by a lack of high-level content and that the quality of PredNet predictions reflected the similarity of consecutive frames.

### Evaluation of low-level feature preservation in humans

After testing artificial systems, we also probed humans on these metamer stimuli to investigate the preservation of low-level and high-level features and how STST and STPS stimuli are perceived. First, we used

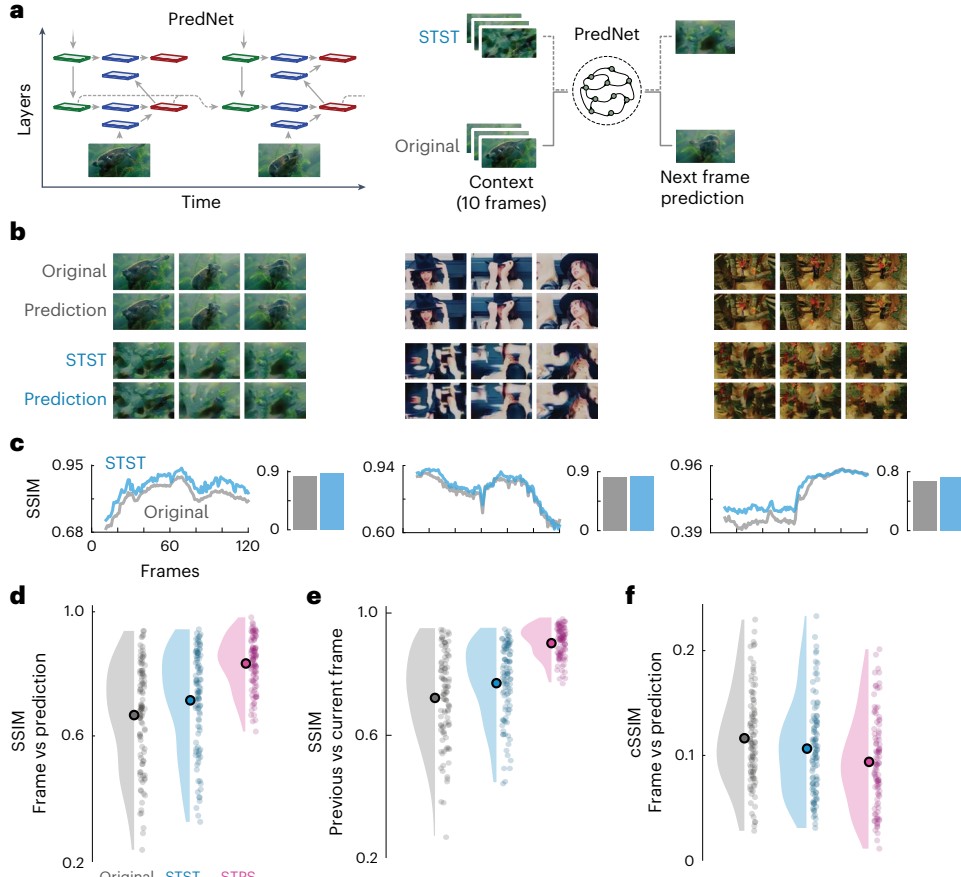

**Fig. 3 | Probing the effect of high-level information in predictive coding networks using STST metamers. a**, Graphical outline of the analysis pipeline. Left: PredNet architecture with input and prediction (blue), representation (green) and error (red) modules. Gray arrows denote information flow, with solid lines indicating forward and feedback flow, and dashed lines indicating recurrent connections. Middle: we passed as inputs to PredNet STST metamer stimuli and their natural counterparts (Original). We separately computed next-frame predictions for both inputs. **b**, Example ground-truth and predicted frames. **c**, SSIM between PredNet predictions and ground-truth frames for original (gray) and STST (azure) stimuli. Bar plots depict video averages. **d**, Raincloud plots showing SSIM between predicted and ground-truth frames for all original, STST and STPS videos of the validation dataset. **e**, SSIM between consecutive video frames for all original, STST and STPS videos of the validation dataset. **f**, cSSIM between predicted and ground-truth frames for all original, STST and STPS videos of the validation dataset. Circles indicate mean values of the distribution. Dots indicate individual videos ($n = 100$). Natural images in **a** and **b** adapted from Google LLC under a Creative Commons license CC BY 4.0.

a video-captioning task (Fig. 4a) in which human participants were asked to provide a text description of the presented videos. We used the same three high-quality videos as used in the previous analyses, alongside their scrambled counterparts. We used deep language models to extract sentence embedding from these captions and performed classification analysis (Fig. 4b) to investigate the discriminability of these descriptions between conditions.

We found that for each of the three videos, the original video captions were significantly discriminable from both STST (mean accuracy = 0.964, all $P < 0.0001$) and STPS (mean accuracy = 0.976, all $P < 0.0001$) descriptions. No discriminability was observed between STST and STPS (mean accuracy = 0.532, all $P = 0.11$). Crucially, we also found that the cosine distance between sentence embeddings for all pairs of original videos were significantly larger (Fig. 4c) than the cosine distances for STST ($P < 0.0001$, $d = 2.84$) and STPS ($P < 0.0001$, $d = 3.22$) video pairs. In other words, the captions of the three original videos were consistent across subjects and different between videos, whereas captions were significantly less distinct for STST and STPS (compare also Fig. 4d). We also observed that the cosine distance between the original and STST embeddings was significantly smaller than between the original and STPS (Fig. 4c, $P = 0.0072$, $d = 0.83$).

A projection of the sentence embedding structure via the uniform manifold approximation and projection (UMAP) dimensionality

reduction[66] method (Fig. 4d) unraveled how the metamer stimuli were perceived substantially differently from the original videos. Furthermore, the actual text captions of the metamers lacked all the semantic references to the object-level features that were present in the original video captions. Together, these results suggest that no high-level content was detected by humans in the STST videos and that the STST videos were described in a more similar way to the natural videos than to the STPS videos.

To further compare the STST and STPS videos, we collected data on human participants in a two alternative forced choice (2AFC) perceptual similarity task (Fig. 4e). Participants were instructed to report which of two sequentially presented option videos (either STST or STPS) were more similar to a reference video (original), shown at the start of each trial. We used the same three high-quality videos as were used in the previous analyses and tasks, alongside their manipulated counterparts. Crucially, when confronted with the decision to judge the similarity of the STST and STPS videos to the natural videos, human observers significantly preferred the STST videos (Fig. 4f; condition 1: preference score for STST = 0.974, $P < 0.0001$). We also conducted control analyses on whether humans recognized that the STST or STPS videos generated from the reference (termed 'A' videos) were more similar to the reference than videos generated from another video (termed 'B' videos) (conditions 2 to 5). We found that, in all control

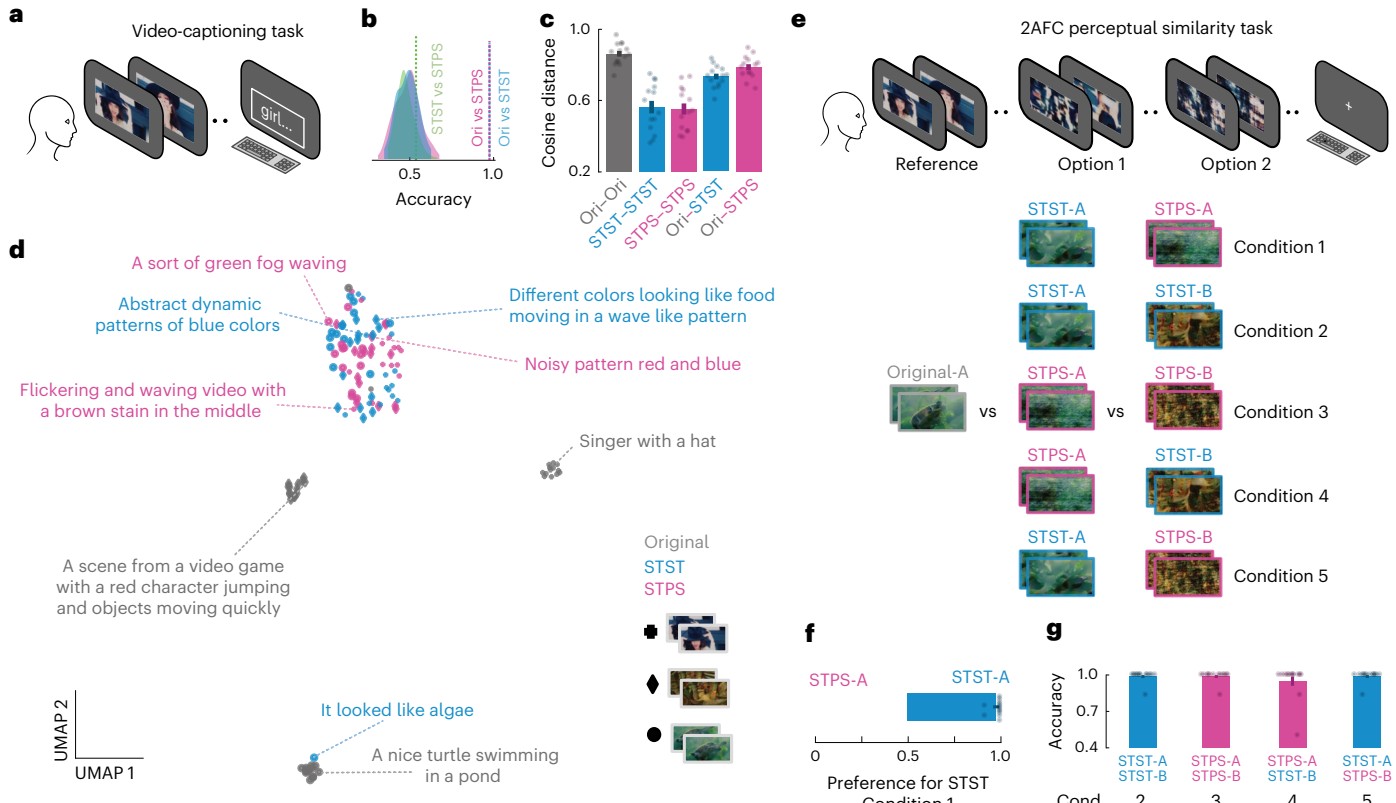

**Fig. 4 | Probing humans on STST and STPS stimuli. a**, Graphical outline of the video-captioning task. **b**, Classification analyses on the sentence embeddings between original, STST and STPS video captions. The density plot represents the null distribution, and vertical dotted lines indicate the observed accuracy pooled across all stimuli. **c**, Cosine distance scores between all pairs of original, STST and STPS videos, as well as between the original video captions and the captions of the STST and STPS stimuli. Error bars indicate s.e.m. Dots indicate human participants (n = 14). **d**, UMAP representation of the sentence embeddings

from all participants and stimuli. Some actual text captions are provided next to the corresponding samples. **e**, Graphical description of the 2AFC perceptual similarity task and all five conditions used to investigate how humans rated the STST and STPS videos. **f**, Preference for STST videos compared to STPS. Error bars indicate s.e.m. Dots indicate individual participants (n = 13). **g**, Accuracy of similarity judgments for the four control conditions. Error bars indicate s.e.m. Dots indicate human participants (n = 13). Natural images in **a**, **d** and **e** adapted from Google LLC under a Creative Commons license CC BY 4.0.

conditions, humans were able to appropriately select the correct video significantly more often than by chance (Fig. 4g, all accuracies > 0.948, all P < 0.0001). Together, these findings show that STST-generated metamers are perceived by humans as perceptually more similar to the original videos than the STPS metamers, which probably reflects the better preservation of low-level spatiotemporal features.

**Model metamers for spatiotemporal factorization**

The two-stream architecture of the network implementing the STST algorithm allows us to independently synthesize spatial and temporal features, a process that can be conceived as spatiotemporal factorization. This provides a unique facility to generate stimuli that are specifically tailored to investigate how spatial and temporal features are processes in visual systems. Here we showcase a version of this application in which we match texture loss in both spatial and temporal streams, but using two separate videos (Fig. 5a). The idea is to generate a video that is matched in space to one video and in time to another. All other hyperparameters were kept the same as for the previously generated model metamers.

Figure 5b presents an example of generated frames alongside their low-level statistics. The spatiotemporally factorized metamer video (blue) is following the spatial statistics of the spatial target video (red) and the temporal statistics of the temporal target video (green). We used this approach in a proof-of-principle study to investigate how biological and artificial visual systems represent spatial and temporal features in these mixed stimuli compared to their natural counterparts.

Specifically, we asked to what extent similarities can be independently judged in the temporal and spatial domains. We hypothesized that, if this were the case, the same spatiotemporally mixed metamer video should be perceived as more similar to either the spatial or temporal natural target video depending on the relevant feature domain.

We performed this experiment on two sets of three videos (see Supplementary Video 2 for the first set and Supplementary Video 3 for the second set). The first stimulus set were the three example videos we also used in all the other experiments (Fig. 5c, top row). We quantified the videos' color distribution with their palette and the optical flow distribution (Fig. 5c). It turned out that, by chance, the three videos in stimulus set 1 had a similar temporal profile and a dissimilar spatial profile. To quantify this, we computed distance measures between each pair of videos along all the low-level statistics we had used so far, with the addition of the full color distribution (Fig. 5d). Although the spatiotemporal similarities were not concerning in all the previous investigations, for the current proof-of-principle experiment this could bias similarity judgments towards the more distinct spatial domain.

We thus implemented a second stimulus set with three videos that showed similarity patterns opposite those of set 1 (Fig. 5c, bottom row). These videos consisted of a quasi-static monkey, a rugby game with the camera following players moving all to the left, and a pole vault athlete running to the right followed by the camera. All videos had a green background. This stimulus set was characterized by more different temporal features and more similar spatial features when compared to set 1 (Fig. 5d). The STST algorithm allowed us to spatiotemporally

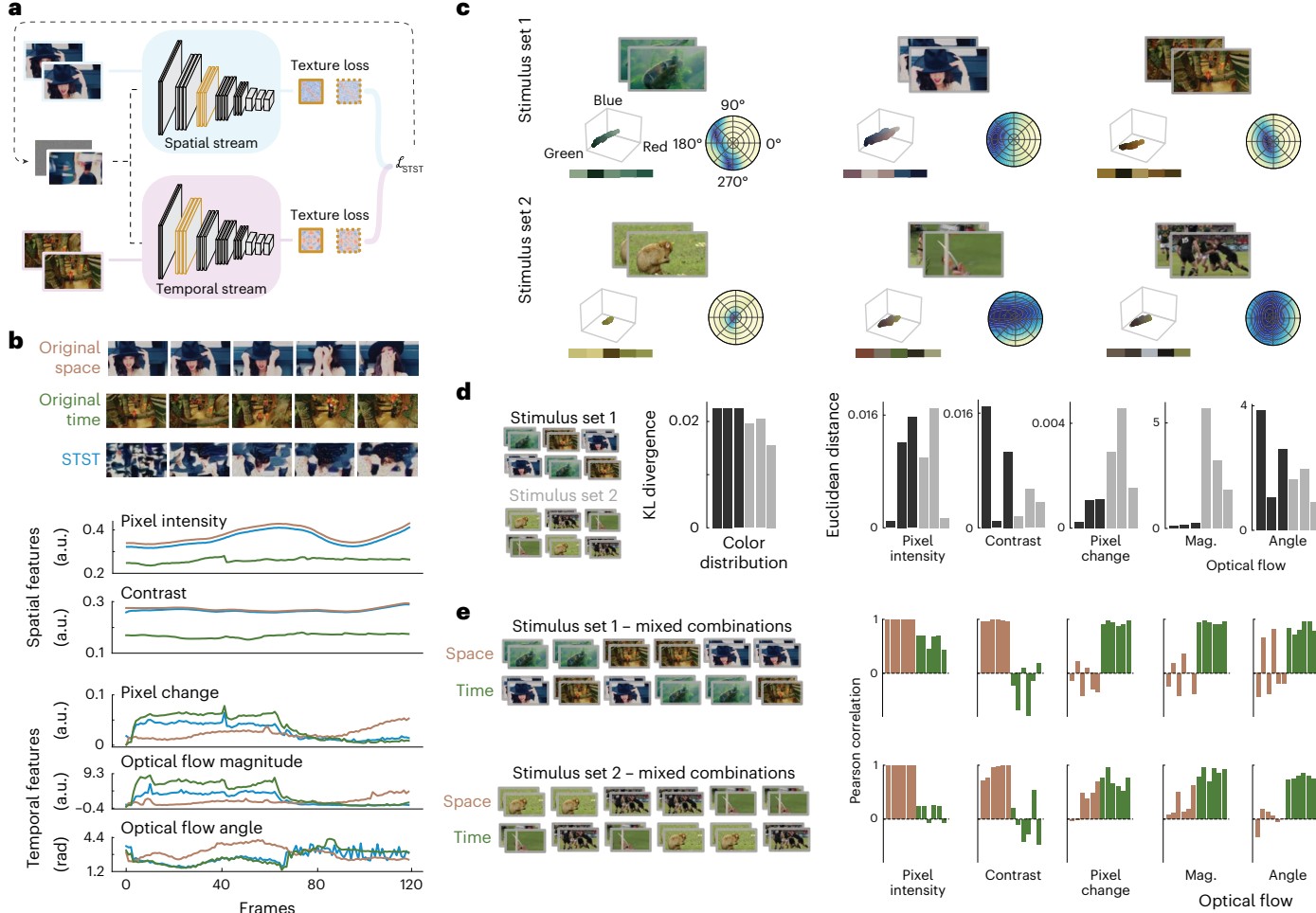

**Fig. 5 | Metamer stimuli for spatiotemporal factorization. a**, Graphical illustration of the STST algorithm for generating a video that is matched with two target videos that are independently given as input to the spatial and temporal streams. **b**, Example of a video generated with this procedure. At the top are frames from the spatial (orange) and temporal (green) target, as well as from the generated STST (azure) video. Below are the low-level statistics of the three videos. **c**, Stimulus properties from stimulus sets 1 and 2. On the bottom left of each video are the 3D color distribution and its palette. On the bottom right, a radar plot shows the distribution of optical flow across the frames. **d**, Distances between all pairs of stimuli in each set for all low-level statistics. For the color

distribution, the distance metric is the KL divergence. For the other low-level statistics we used the Euclidean distance. The legend on the left reports the order of the bar plots on the right. **e**, Pearson correlation between the original video from the spatial (orange) or temporal (green) stream and the corresponding spatiotemporally factorized metamer video for each low-level statistic. Notably, the metamer videos have higher correlation with the spatial targets in the spatial statistics and higher correlation with the temporal targets in the temporal statistics. The legend on the left reports the order of the bar plots on the right. Mag., magnitude of the optical flow. Natural images in **a**–**e** adapted from Google LLC under a Creative Commons license CC BY 4.0.

mix the target videos of both stimulus sets. For both stimulus sets, the spatial and temporal features of the six mixed metamer videos were correlated with either the spatial (average correlation in stimulus sets 1/2, $r = 0.98/0.95$) or temporal (average correlation in stimulus sets 1/2, $r = 0.90/0.78$) natural target video, respectively (Fig. 5e).

**Spatial bias in humans and models with factorized metamers**
Based on these stimuli, we collected data from human participants using a 2AFC spatiotemporal perceptual similarity task (Fig. 6a). Participants were instructed to report which of two sequentially presented option videos (either the original spatial or temporal target) appeared more similar to the previously presented reference video (one of the mixing combinations). Before the start of each trial, a text cue instructed participants to base their similarity judgments on either spatial or temporal features. We quantified the results as the accuracy of human observers in detecting the correct spatial or temporal target video for the feature domain at hand.

As expected, for videos with the same spatial and temporal target videos (matrix diagonal in Fig. 6b,c), participants were significantly

better than chance at discriminating spatial and temporal similarity (all $P < 0.0002$, all $d > 2.29$). However, these trials were comparatively easy, as the comparison was made between the original target video and another original video. Accordingly, we next focused on the mixed metamer stimuli, which matched both options in either feature dimension (off-diagonal in Fig. 6b,c). We found that for these challenging trials too, the spatial accuracy was higher than chance for both stimulus sets (stimulus set 1/2: accuracy = 0.88/0.89, $P < 0.0001/0.0001$, $d = 3.82/3.72$). Also, for these trials, the temporal accuracy was higher than chance for the temporally more distinct stimulus set 2, but not for set 1 (stimulus set 1/2: accuracy = 0.42/0.67, $P = 0.387/0.039$, $d = -0.35/0.96$). For both sets, the spatial accuracy was significantly higher than the temporal one (stimulus set 1/2: $P = 0.001/0.013$, $d = 1.92/1.09$). These results show that humans can independently judge spatial and temporal video similarities, and suggest a similarity judgment bias towards spatial features.

Finally, we also tested deep vision models trained for video classification (ResNet18-3D and ResNet18-MC3) using the spatiotemporally factorized videos (Fig. 6d). To compare their behavior with

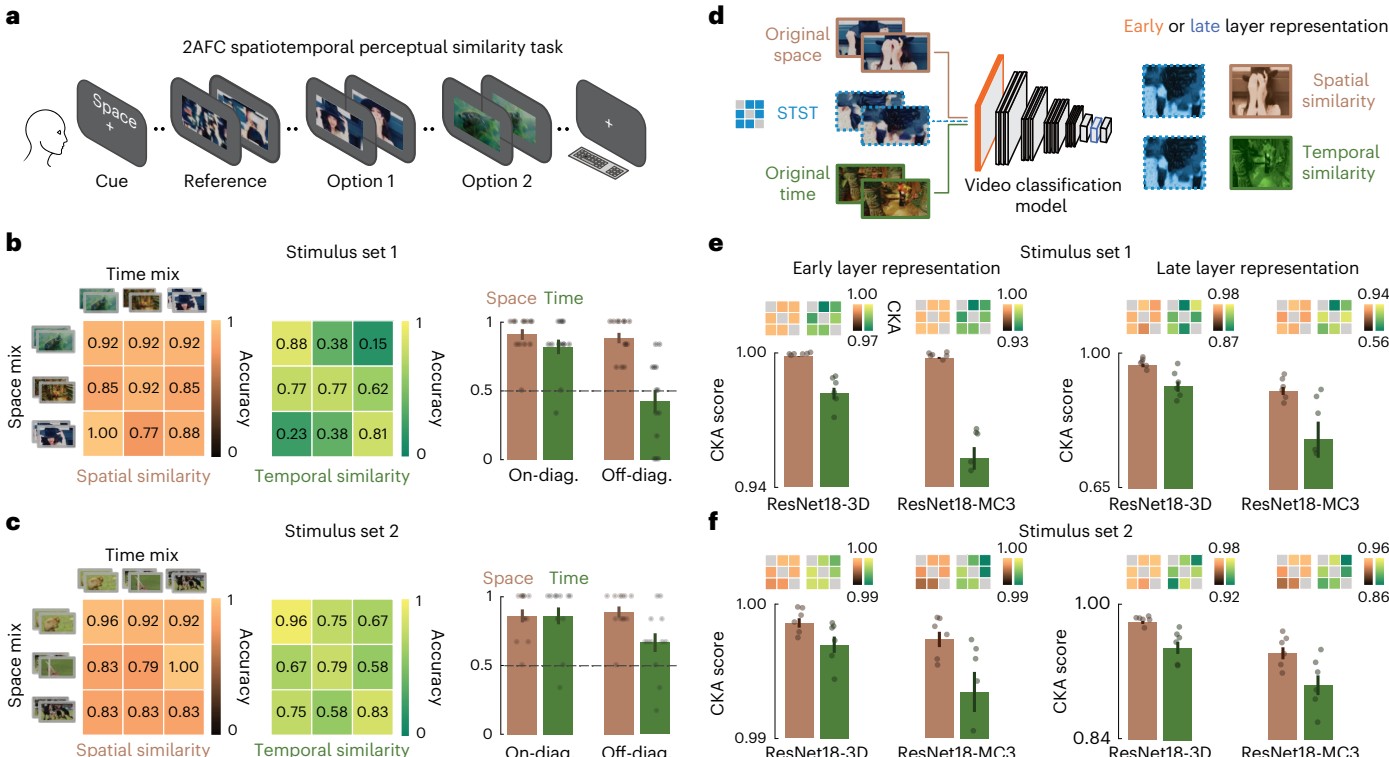

**Fig. 6 | Testing humans and deep video models on spatiotemporally factorized metamer stimuli. a**, Graphical description of the 2AFC spatiotemporal perceptual similarity task performed by human participants. **b**, Spatiotemporal discriminability for stimulus set 1 ($n = 13$). Mixing matrices show the accuracy of participants in discriminating either space (orange) or time (green) for all combinations. Bars show average accuracies across on-diagonal and off-diagonal entries. Error bars represent s.e.m. Dashed horizontal lines indicate chance level. Dots indicate individual participants. **c**, Spatiotemporal discriminability for stimulus set 2 ($n = 12$). The conventions are the same as in **b**. **d**, Graphical illustration of the methodology employed to test deep video models in analogy

to the behavioral task for humans. All off-diagonal STST mixing combinations were given as input to the models alongside their respective spatial and temporal targets. We then computed the CKA score in early and late layers between the STST and original videos to assess either the spatial (orange) or temporal (green) representational similarity. **e**, CKA results for stimulus set 1. Bars show the average CKA scores across all off-diagonal combinations. Error bars denote s.e.m. Dots indicate individual combinations ($n = 6$). The corresponding mixing matrices are shown above the bars. **f**, CKA results for stimulus set 2. The conventions are the same as in **e**. Natural images in **a**–**d** adapted from Google LLC under a Creative Commons license CC BY 4.0.

the human data, we passed as inputs all off-diagonal mixed metamer videos and the corresponding spatial and temporal target videos. We extracted the models' early and late layer representations and computed the CKA score between the STST videos and their original videos to obtain a spatial and temporal similarity score. Across both stimulus sets (Fig. 6e,f), we found that the models' representations of the mixed metamer videos were more similar to the spatial target than to the temporal target. We observed this pattern for both early and late layer representations and across both tested models. Overall, these results suggest that human similarity judgments and representational similarities in deep vision models trained for video classification are biased towards spatial over temporal features, providing insights into the representational capabilities of biological and artificial systems for the spatiotemporal integration of dynamic visual information.

## Discussion

Here we propose an STST algorithm as a flexible framework for dynamic visual stimulus generation. The proposed framework may facilitate vision science beyond the applications shown here. For instance, in the present examples we only manipulate the texture (style) of the spatiotemporal targets, either from the same natural video or by factorizing space and time with two different videos. Future applications of STST may explore all possible combinations of content and texture loss along the spatial and temporal dimensions, with up to four distinct target videos. For example, one other application is the

so-called video style transfer[67–69], in which the content from one target video is combined with the style from another video or image. For this application, other algorithms have been proposed, some of which also exploit optical flow information to improve perceptual stability[68]. The proposed STST algorithm differs from these video style transfer alternatives[67–69] in terms of model architecture (two-stream approach), flexibility and its ability to match spatiotemporal stimulus statistics at various levels. These aspects are especially relevant when conceiving our approach as a dynamic visual stimulus generation framework. This is particularly the case in the field of NeuroAI, where the comparison of biological and artificial systems is increasingly evaluated by means of out-of-distribution stimuli[29,70–72], that is, stimuli that diverge from the training set of artificial systems and that are faced by biological systems during their lifetimes. The STST framework thus opens an intriguing out-of-distribution regime for video stimuli.

Our specific choice of spatial and temporal stream models was motivated by previous studies on image style transfer and dynamic texture synthesis[21,22,33]. Beyond these specific models, the STST algorithm provides a general framework for spatiotemporal factorization. Future research may explore different settings of the proposed framework, for example by using different spatial models or temporal stream models that allow the consideration of longer temporal scales than with the model implemented here. Nevertheless, using the described parameters and models, the proposed framework may already provide a suitable and robust solution for many applications, including the showcased examples.

As such examples, we generated metamer stimuli that were invariant to the spatial and temporal texture of natural videos. We benchmarked our method to the only other currently available method (to the best of our knowledge), STPS[50], to generate dynamic stimuli preserving low-level spatiotemporal statistics. STST outperformed STPS, especially in preserving the optical flow of videos. This may be ascribed to the fact that STPS randomizes the phase of the spatiotemporal frequency spectrum, whereas our method uses, as part of its objective, the Gram matrix of the layer activations of an optical flow estimation model. These findings highlight the potential of our algorithm for the study of object recognition in dynamic natural vision.

We tested the effectiveness of STST-generated stimuli in preserving low-level features and ablating high-level features using state-of-the-art deep vision models[9,10] for image and video classification. These models parallel the hierarchical organization of the mammalian visual cortex in the sequential extraction and integration of visual features[41,42,52,73]. Notably, we employed convolutional-only models in our analysis, as it has been shown that vision transformer architectures[74,75] do not exhibit a hierarchical structure in the visual feature extraction process[76].

Our results show that not only image, but also video classification models exhibit similar early-layer activation for metamer and natural stimuli, whereas later layers diverge. Notably, there is less research on the effective nature of hidden representations of video-processing models and how they relate to biological visual systems[77,78]. Our results confirm that for video models and low-level temporal features also, early layers similarly encoded metamer and natural videos, and late layers are also disrupted in metamers when integrating motion.

We also investigated the representational capabilities of PredNet, a DNN inspired by the concepts of predictive coding[58–60]. The measures of predictiveness quantified its predictive ability beyond static image quality or simple frame-copying[64], uncovering limitations in exploiting the object-related motion patterns present in natural videos[58,64]. We speculate that this effect may be due to the fact that we used the version of PredNet trained to minimize only the lowest-level prediction error, as this version has been shown to perform best on next-frame prediction tasks[60,64]. However, this architectural implementation differs from the original theoretical proposition of predictive coding[58]. Furthermore, our analyses revealed that STST better preserved consecutive frame dependencies than STPS, which renders STST particularly well suited for this application.

Complementarily to artificial systems, we also investigated the effectiveness of our algorithm in human observers. Human participants described metamer stimuli less consistently than natural videos, reflecting their lack of high-level content. Crucially, descriptions and similarity ratings indicated that STST metamer videos better preserved low-level features compared to STPS, with a strong preference for STST in direct comparisons to natural videos. This preference probably stems from STST's better match with natural optical flow compared to STPS, suggesting that optical flow is important for perceived naturalness in dynamic stimuli. Consistently, deep vision models showed greater similarity between STST and natural videos than with STPS, highlighting a notable parallel between biological and artificial systems in the role of optical flow in shaping representations[79], even in models trained solely on image classification.

Finally, we introduced a procedure for the spatiotemporal factorization of dynamic stimuli by matching the model layer activations of the spatial and temporal streams to two different target videos. Our results suggest a spatial bias in similarity judgments, which may stem from the additional computational effort due to temporal integration associated with temporal information[80]. Alternately, and well compatible with our findings in deep vision models, this bias may reflect a quantitative bias in the number of computational units for spatial as compared to temporal features. Further investigations are required to investigate these alternatives. A possible confounding factor of

these findings is the relative discrepancy between metamer and target videos for spatial and temporal features. Specifically, in factorized metamers, spatial features were slightly better matched with their natural targets than temporal features. Moreover, the average correlation of temporal features with their targets was higher in stimulus set 1 (0.90) than in stimulus set 2 (0.78). Nevertheless, in the temporal task, both humans and the deep vision model had higher accuracy and CKA scores for stimulus set 2 than for set 1. This provides evidence against the hypothesis that the discrepancy between metamer and target features explains the results.

In conclusion, the proposed STST algorithm allows the powerful manipulation and synthesis of video stimuli for vision research. Our results shed light on potential applications of STST and promote it as a versatile tool for dynamic stimulus generation.

## Methods
### Model architecture
Our proposed algorithm is based on a two-stream model architecture consisting of one module (or 'stream') acting as a spatial feature detector on each frame of the target video, and another module performing feature detection in the temporal domain across consecutive pairs of frames (Fig. 1). The model is agnostic to the specific implementations of both spatial and temporal modules, so any pretrained differentiable model can be used. Thus, in the following, we formally denote the spatial module as a differentiable function $\mathcal{S}(x_t)$ that takes as input the current frame defined as a 3D tensor $x_t \in \mathbb{R}^{H \times W \times C}$, where $H$ and $W$ are the height and width of the image in pixels, $C$ is the number of channels in the frame (in our case $C = 3$, as we are working with RGB-encoded frames) and the subscript $t$ denotes the time step. Each element of the tensor $x_t$ represents the intensity of the pixel at position $(H, W)$ in channel $C$, with a value in the range [0, 1]. We also denote the temporal module as a differentiable function $\mathcal{T}(x_t, x_{t-1})$ that takes as input both the current ($x_t$) and previous ($x_{t-1}$) frame.

We opted to maintain consistency with previous works on NST and its extension to dynamic textures. Accordingly, we adopted VGG-19 (ref. 43) as the spatial module and the MSOE model[33,44] as the temporal module. VGG-19 is a pretrained CNN model[43] that was trained on the ImageNet1k dataset[81,82] for object recognition, and it was used originally in both ref. 21 and ref. 33. We also opted for this model because recent evidence has shown that this type of architecture outperforms many other models in the task of style transfer or texturization[83], although the reasons for this remain debated[46,83,84]. MSOE is a pretrained CNN model that was trained on the UCF101 dataset[85] to predict optical flow and was similarly used in ref. 33 as the temporal module. We opted for this model because it is generally invariant to spatial features[33] and thus relies entirely on the temporal dynamics of the video data to estimate optical flow.

We denoted the layer activations of the CNN models as $\mathcal{A}^\ell \in \mathbb{R}^{N_\ell \times M_\ell}$, where $N_\ell$ and $M_\ell$ refer to the number of filters and the number of spatial locations (the height times the width of the feature map) in convolutional layer $\ell$. The layer activation $\mathcal{A}^\ell$ is defined as the results of the forward pass to that specific layer $\ell$ after both the convolutional operator and the nonlinear activation function. Here, note that we refer to the layer activation $\mathcal{A}^\ell$ for both the forward pass of $\mathcal{S}(x_t)$ and $\mathcal{T}(x_t, x_{t-1})$, indiscriminately.

### Optimization procedure
We now define the optimization procedure to generate what we refer to as 'model metamers', which are generated dynamic stimuli possessing certain spatiotemporal features that are indistinguishable from their natural counterparts for the two-stream model[34]. The STST algorithm can generate metamers that are matched in terms of content (shape) and style (texture) along both spatial and temporal streams. Following the work of Gatys and colleagues[21], we define the loss function of the content-matching procedure as the squared difference

between the layer activations $\mathscr{A}^\ell$ of the target frame $x_t$ and the layer activations $\hat{\mathscr{A}}^\ell$ of the generated frame $g_t \in \mathbb{R}^{H \times W \times C}$:

$$\mathcal{L}^\ell_{\text{content}}(x_t, g_t) = \frac{1}{2N_\ell M_\ell} \sum_{i=1}^{N_\ell} \sum_{j=1}^{M_\ell} \left( \mathscr{A}^\ell_{i,j} - \hat{\mathscr{A}}^\ell_{i,j} \right)^2 \tag{1}$$

Optimizing this loss function for all frames allows the algorithm to match the spatial or temporal structure of the target video. In other words, by using merely the content-matching procedure, the algorithm generates approximately the same video that was used as target. Note that, for the temporal module $\mathscr{T}(x_t, x_{t-1})$, the input is both the current $x_t$ and previous $x_{t-1}$ target frames, so the current generated frame $g_t$ and the previous generated frame $g_{t-1}$ also need to be provided.

Conversely, to perform texture matching we need to first obtain a representation of the texture information[21]. Here we refer to texture as either spatial or temporal recurrent patterns (namely, a set of stationary statistics). We define this texture representation as the correlation between different filter responses within a layer activation[21,33], encapsulated by the so-called Gram matrix $G^\ell \in \mathbb{R}^{N_\ell \times N_\ell}$, whose entries are given by

$$G^\ell_{i,j} = \frac{1}{N_\ell M_\ell} \sum_{k=1}^{M_\ell} \mathscr{A}^\ell_{i,k} \mathscr{A}^\ell_{j,k} \tag{2}$$

Thus, the loss function of the texture-matching procedure is defined as

$$\mathcal{L}^\ell_{\text{texture}}(x_t, g_t) = \frac{1}{2N_\ell^2} \sum_{i=1}^{N_\ell} \sum_{j=1}^{N_\ell} \left( G^\ell_{i,j} - \hat{G}^\ell_{i,j} \right)^2 \tag{3}$$

where $G^\ell$ is the Gram matrix computed from layer activations $\mathscr{A}^\ell$, and $\hat{G}^\ell$ is computed from $\hat{\mathscr{A}}^\ell$. Generating metamers with the $\mathcal{L}^\ell_{\text{texture}}$ loss results in videos that no longer maintain a retinotopic correspondence with the original natural video. However, a variety of low-level statistical properties are preserved, depending on which dimension it is applied to. For example, in the spatial stream, the luminosity and contrast are well preserved, as is the magnitude of the optical flow or the global motion in the temporal stream. We can finally define the general form of the loss function for our STST algorithm as

$$\mathcal{L}_{\text{spatial}}(x_t, g_t) = \sum_{\ell=1}^{L_s} \left( \alpha \mathcal{L}^\ell_{\text{sc}}(x_t, g_t) + \beta \mathcal{L}^\ell_{\text{st}}(x_t, g_t) \right) \tag{4}$$

$$\mathcal{L}_{\text{temporal}}(x_t, g_t, x_{t-1}, g_{t-1})$$
$$= \sum_{\ell=1}^{L_t} \left( \theta \mathcal{L}^\ell_{\text{tc}}(x_t, g_t, x_{t-1}, g_{t-1}) + \lambda \mathcal{L}^\ell_{\text{tt}}(x_t, g_t, x_{t-1}, g_{t-1}) \right) \tag{5}$$

$$\mathcal{L}_{\text{STST}}(x_t, g_t, x_{t-1}, g_{t-1}) = \mathcal{L}_{\text{spatial}}(x_t, g_t) + \mathcal{L}_{\text{temporal}}(x_t, g_t, x_{t-1}, g_{t-1}) \tag{6}$$

where $L_s$ and $L_t$ are the number of selected layers in the spatial and temporal modules, respectively. Moreover, $\alpha$ and $\beta$ represent the weights associated with the content $\mathcal{L}^\ell_{\text{sc}}$ and texture $\mathcal{L}^\ell_{\text{st}}$ loss in the spatial loss $\mathcal{L}_{\text{spatial}}$, and $\theta$ and $\lambda$ are the weights for the content $\mathcal{L}^\ell_{\text{tc}}$ and texture $\mathcal{L}^\ell_{\text{tt}}$ loss in the temporal loss $\mathcal{L}_{\text{temporal}}$. These weights are set as hyperparameters that modulate the relative contribution of each of the four terms to the general loss function $\mathcal{L}_{\text{STST}}$. Furthermore, the selection of a restricted number of terms (for instance, selecting only the spatial and temporal texture losses) can be seen as setting the undesired terms' weights to zero. This provides a versatile tool for dynamic stimulus generation, as one can perform many combinations of the four losses that compose the general loss function. Crucially, one can select more than one target video and even assign a different target video for each of the four loss terms.

Once the total loss for the optimization procedure is defined, the actual update of the parameter space $g_t$ consists of applying a gradient-based optimization method such as gradient descent, which iteratively adjusts the parameters in the direction that minimizes the loss:

$$g_t := g_t - \eta \frac{\partial \mathcal{L}_{\text{STST}}}{\partial g_t} \tag{7}$$

where $\eta$ is the learning rate that weights the gradient of the loss with respect to the parameters and is set as a hyperparameter. The number of iterations needed to update the parameters $g_t$ is also a hyperparameter of STST. Importantly, the gradients are computed and applied only with respect to the current generated frame, even in the case in which the temporal module is performing the forward pass with both current and previous frames. This is because optimizing the full pair of consecutive frames would lead to instabilities in the optimization. The generation of the dynamic stimulus is finished when the optimization procedure has been iterated across all frames.

## Perceptual stabilization via preconditioning

Although the optimization procedure described above performs well for videos with simple spatiotemporal dynamics, as similarly shown by Tesfaldet and colleagues[33] for the generation of dynamic textures, it encounters substantial limitations when applied to natural video sequences. Empirically, it tends to manifest considerable perceptual instabilities both in the space and time domains, so we incorporate a family of techniques for image and video synthesis that substantially improve its robustness and perceptual stability. The employed techniques can be generally described as a 'preconditioning' of the optimization procedure. In mathematical optimization, preconditioning refers to a transformation of an optimization problem to condition it to a form that is more suitable for finding optimal solutions. Intuitively, preconditioning changes the basins of attraction, allowing the optimization process to approach preferred solutions more easily[46].

The first preconditioning technique that we introduced is the addition of the total variation (TV) loss[45] to our loss function. The TV loss is a regularization term often used in image-processing algorithms to encourage smoothness in the output while preserving edges, and it was introduced in the literature of image generation from early works on the DeepDream algorithm[46,86]. Intuitively, the idea is that in natural images, pixel values tend to change gradually except at the edges, where there are abrupt changes. The TV loss penalizes the sum of the absolute differences between neighboring pixel values, as described by the following anisotropic 2D version of the formula:

$$\mathcal{L}_{\text{TV}}(g_t) = \frac{1}{HWC} \sum_{k=1}^{C} \sum_{i,j}^{H,W} \left| g_t^{i+1,j,k} - g_t^{i,j,k} \right| + \left| g_t^{i,j+1,k} - g_t^{i,j,k} \right| \tag{8}$$

This additional loss helps to prevent the excessive high-frequency noise that results from the optimization procedure described above. Specifically, this arises from the spatial module and most probably the convolutional nature of the selected model, as it has been shown that strided convolutions and pooling operations can create high-frequency patterns in the gradients[46,84]. Note that this loss term is applied on the current generated frame during the optimization process, so there is an additional hyperparameter $\omega$ to weight the contribution of the $\mathcal{L}_{\text{TV}}$ term to the total loss.

Another technique we borrowed from the DeepDream field is the spatial multiscale approach[14,46]. It basically consists of starting the optimization with an image at a lower resolution and iteratively enlarging it to the original size. This is done to inject spatial frequency biases into the parameter space, by initially providing low-frequency patterns and then refining high-frequency ones. We implemented this approach by resizing both the targets and generated frames using bilinear

interpolation and a hyperparameter $\sigma$ to control the so-called octave scale. The octave values $o$ are the exponent power of the octave scale, used for multiplying them with the original frame size $H \times W$, for example, $H_o = H o^o$ and $W_o = W o^o$. Note that $o = 0$ yields the original resolution. The aforementioned optimization procedure is applied on each octave, possibly even with different hyperparameters, such as different learning rates for different octaves. We followed the authors that introduced the multiscale approach and normalized the gradients of the total loss with respect to $g_t$ by dividing them with their standard deviation across the height $H_o$ and width $W_o$ dimensions[14,46].

In the STST algorithm we also implemented a method to match the color distribution between $x_t$ and $g_t$. We noticed that this substantially improved the similarity between the targeted and generated color distributions compared to only applying the optimization. Thus, following the multiscale procedure, we applied a color transfer algorithm as a postprocessing step[47,48]. Instead of using a naïve approach such as histogram matching for each channel individually[87], the employed algorithm finds a transformation from the full 3D probability density function of the pixel intensity of the target frame to the generated frame, and also reduces the grain artifacts by preserving the gradient field of the target frame[48].

Finally, we applied a blending operation on the frame transitions to reduce flickering artifacts. This turned out to be a critical step that improved the perceptual stability by preconditioning the initial conditions of the optimization procedure at the current frame with the post-processed frames at the previous time point and the addition of uniform noise $\mu \in \mathbb{R}^{H \times W \times C}$, as follows:

$$g_t \leftarrow \varphi g_{t-1} + (1 - \varphi)\mu, \; \mu \sim \mathcal{U}(0, 1) \tag{9}$$

where $\varphi$ is the hyperparameter that controls the blending ratio and $\leftarrow$ stands for initialization. For instance, setting $\varphi = 0.9$ results in the current frame being initialized with 90% the pixel intensity of the previous frame, with the rest being uniform noise. Although this blending operation substantially improved the temporal stability of our method, the initial frames of the generated stimuli had slightly different low-level statistics compared to the rest of the frames. In other words, the relative stationarity of the low-level statistics in the generated stimuli with respect to the target stimuli started to be consistent only after a few initial frames. We assumed that this was because the very first frame is initialized only as noise, because there is no previous frame with which to precondition. Also, it is difficult to increase the speed of the optimization process as our algorithm needs to have a relatively low value of $\eta$ to maintain numerical stability. This can be interpreted as the search for stable local minima not just within each frame, but across them, as the parameter space of the next frames is preconditioned on the previous ones by the blending operation. Thus, to remedy the first frame effect, we mirror-padded the target and the generated stimuli for the first $\xi$ frames. In other words, we took the first $\xi$ frames of the target stimuli, flipped their order, and concatenated them before the first frame. We then discarded the initial padded frames after the optimization.

### Stimulus generation for object-recognition studies
We generated dynamic stimuli by matching spatiotemporal features to their natural counterparts. We optimized metamer stimuli to match both the spatial and temporal textures of three high-quality video clips collected from the YouTube-8M dataset[49]. The target videos consisted of 120 frames at 60 frames per second (FPS), with a frame resolution of 360 × 640. We also performed validation analyses on 100 video clips collected from the validation set of the Kinetics400 dataset[51]. We selected one random video from 100 different action labels to maximize the diversity in the sample, analyzing the first 60 frames from each video, corresponding on average to 2 s, because the majority of the videos were recorded at 30 FPS. The identity code of each

video in the dataset is provided in Supplementary Table 1. All statistical comparisons performed on this validation dataset were conducted using two-tailed paired $t$-tests and Cohen's $d$ as the effect size metric.

We included in our total loss function only the spatial $\mathcal{L}_{st}^{\ell}$ and temporal $\mathcal{L}_{tt}^{\ell}$ texture loss components to eliminate any high-order regularity. For the spatial stream $\mathcal{S}(x_t)$, we used VGG-19 (ref. 43) and selected the layers conv1_1, conv2_1, conv3_1, conv4_1 and conv5_1 for the texture loss (as in ref. 21), and for the temporal stream $\mathcal{T}(x_t, x_{t-1})$ we used MSOE[33] and selected the concatenation layer (as in ref. 33), where the multiscale feature maps of orientations are stored, for the texture loss. For each frame, we selected three octaves $o \in [-2, -1, 0]$ with an octave scale $\sigma = 1.5$, resulting in frame resolutions of 160 × 284, 240 × 426 and the original one. The weights associated to both texture losses ($\beta$ and $\lambda$) were always set to 1, and the weight $\omega$ for the $\mathcal{L}_{TV}$ loss was set in dependence on the octaves, namely having values of 0.05, 0.1 and 0.5, respectively. The hyperparameters of the optimization process were octave-dependent, with iterations of 250, 750 and 1,000 and the learning rate $\eta$ set as 0.001, 0.003 and 0.005, respectively. We set the blending ratio $\varphi$ to 0.95 and the number of padding frames $\xi$ to 5. Using an NVIDIA A40 graphical processing unit, this optimization process for the three high-quality videos took on average 6 h.

### Comparison to existing methods
We generated dynamic metamer stimuli from the same target videos with a different algorithm[50], STSP. STSP consists of three steps, starting with randomization of the phase spectrum of each frame separately using a 2D fast Fourier transform (FFT). The random phase angles were sampled from a uniform distribution with range $[-\pi, \pi]$, then applied to all frames. Next we used a 3D FFT to also randomize the phase spectrum of the full spatiotemporal data. Finally, we used the same color transfer algorithm[48] as for STST to enable a fair comparison of the methods. We applied the first step to each channel of the frames separately, the second to all frames together but for each channel separately, and the third step to each frame, because it handles the full 3D distribution of pixel intensity.

### Spatiotemporal feature analysis
We computed four basic spatiotemporal features from the original videos, our STST stimuli and the STPS stimuli. For the spatial low-level features, for each frame, we computed the average pixel value across all $H$, $W$ and $C$ dimensions and the luminance contrast as the standard deviation across both $H$ and $W$ (ref. 50), after transforming the image from the RGB color space to grayscale using the dot product between the channel values and the following vector [0.299, 0.587, 0.114]. For the temporal low-level features, we computed the pixel change as the average across all $H$, $W$ and $C$ dimensions of the absolute difference between consecutive frames[50], and the optical flow feature as the average across $H$ and $W$ of the magnitude and the angle of the dense optical flow estimation between consecutive frames (grayscaled as above), using the Farnebäck algorithm[88] with the pyramid scale set to 0.5 with five levels, an averaging window size of 13, ten iterations per level, five pixel neighbors used to find the polynomial expansion in each pixel, and a Gaussian kernel with a standard deviation of 1.1 for smoothing the derivatives used as a basis for the polynomial expansion.

### Testing vision models on metamer stimuli
We tested the effects of STST stimuli on hidden activations of current state-of-the-art deep learning models for image and video classification. Data analyses were performed using Python 3.9, TensorFlow 2.4 and PyTorch 2.2. We used ResNet50 (ref. 54) and ConvNeXt-T[55], trained on the Imagenet1K dataset[81], as image classification models, and ResNet18-3D[56] and ResNet18-MC3 (ref. 57), trained on the Kinetics400 dataset[51], as video classification models. For the video classification models, we used five consecutive frames as

input and progressed throughout the video with a one-frame step size. For image classification models, we processed individual frames. As early layers, we selected the 'maxpool' layer for ResNet50, 'features.0' for ConvNeXt-T and 'layer1.0.conv1.1' for both ResNet18-3D and ResNet18-MC3. As late layers, we selected 'layer4.2.relu_2' for ResNet50, 'features.7.2.add' for ConvNeXt-T and 'layer4.1.relu' for both ResNet18-3D and ResNet18-MC3. This layer nomenclature was adopted from the torchvision[89] implementation of these models. CKA[53] with a linear kernel was used as a similarity score of the models' layer activations between natural and STST stimuli.

### Testing high-level representations in predictive coding networks

We used metamer stimuli to investigate the role of high-level representations in PredNet[60], a deep convolutional recurrent neural network with four hierarchical levels that was trained with self-supervised learning methods with the KITTI dataset[90] to perform next-frame prediction[61]. We used the same implementation as ref. 60 and focused our analyses on the model that had the highest predictive performance[60,64], namely the one that was trained to minimize the prediction error from the lowest level of the hierarchical structure of PredNet (in the original paper this was referred to as PredNet $L_0$). We passed to the model the metamer stimuli alongside their natural counterparts in a sliding-window fashion, using as context the last ten frames, in line with the original training hyperparameter[60]. The inputs were preprocessed by rescaling them to a frame resolution of $128 \times 160$ as required by the model. We extracted from the model the predicted next frame and the average prediction error coming from the output of the error module across all levels of the hierarchy. We then computed the SSIM[65] and its dynamic version conditional on the previous frame (cSSIM)[64]. cSSIM measures how different the predictions are from the previous frame, quantifying how risky the prediction of the model is in comparison to simply performing a copying of the previous frame as a prediction for the current one:

$$\text{cSSIM}\left(f_t, f_{t-1}, \hat{f}_t\right) = \left(1 - \text{SSIM}\left(f_{t-1}, \hat{f}_t\right)\right) \text{SSIM}\left(f_t, \hat{f}_t\right) \quad (10)$$

where $f_t$ denotes the current frame (either from the original or STST video) and $\hat{f}_t$ represents PredNet's prediction of the current frame.

### Testing model metamers for object recognition in the human visual system

We conducted behavioral experiments to investigate the effects of our metamer stimuli for object recognition on human participants. The experiments were conducted in accordance with the Declaration of Helsinki and approved by the ethics committee of the University of Tübingen. All subjects gave informed consent. No statistical method was used to predetermine the sample sizes of the human experiments. Sample sizes were chosen in accordance with typical sample sizes used in similar studies within the field, ensuring consistency and comparability with established research practices. No data were excluded from the analyses.

First, we collected data for the video-captioning task ($n = 14$, mean age = 30.4 years (3.1 s.d.), six females). We asked participants to type on a keyboard an English text description of the presented videos, which had a visual angle of 15°. We instructed them to describe the video as they would describe it to a person that had not seen the video. We restricted them to use between 2 and 30 words and use only letters and spaces, without punctuation. Each trial consisted of the video playing followed by a box in which participants inserted their caption. After an inter-trial interval (ITI) of 2 s, the next trial started. We used the same three high-quality videos we used in the previous analyses, alongside their STST and STPS counterparts, for a total of nine videos. The order of appearance was pseudo-randomized by counterbalancing across participants whether they started to watch the STST followed by the

STPS and original or starting with STPS and then the STST and original. We opted for this design to avoid the presentation of the original videos first, which could have biased the subsequent trials.

We used the Sentence Transformers library (version 3.0.1)[91] with the paraphrase-mpnet-base-v2 deep transformer model, a Siamese BERT model pretrained for the specific purpose of clustering and semantic search, to extract 768D sentence embeddings from these captions. These embeddings were computed as the mean pooling of the last layer output, taking into account the attention matrix for correct averaging. Next, we performed classification analysis using the support vector machine model with the radial basis function as kernel (rbf-SVM), a regularization value $C$ of 1 and a three-fold-stratified cross-validation scheme. The classification was performed for every combination of original, STST and STPS videos on each stimulus, then the results were averaged across stimuli. The statistical significance of the results was tested by randomly shuffling the class labels and generating a null distribution of 10,000 samples of model performance, which was evaluated with the accuracy score. We also computed the cosine distance between the sentence embeddings of all pairs of original, STST and STPS video descriptions and between the original videos and either their respective STST or STPS counterpart videos. Statistical comparisons were evaluated using two-tailed paired $t$-tests and Cohen's $d$ as the effect size metric. We also visualized the sentence embeddings using UMAP, with 15 neighbors, cosine distance as a metric, 200 epochs, a learning rate of 1, a minimum distance parameter of 0.1, the spread set to 1 and spectral initialization.

Second, we collected data for the 2AFC perceptual similarity task ($n = 13$, mean age = 30.0 (2.8 s.d.), six females). Six participants completed the task after the previous video-captioning task. Each trial consisted of three videos, interleaved by 300 ms and presented with 15° visual angle. The first video was the reference, and the second and third the options of the 2AFC. The videos were the same as the videos used in the previous task, so nine in total (three originals and their STST and STPS counterparts). Participants were instructed to report which of the two option videos (either STST or STPS) were more similar to the reference video (original) by pressing the D and L keys to select the first and second option, respectively. To provide a definition for video similarity, the subjects were instructed as follows: 'There are no right or wrong answers in this task. We are interested in your personal judgment. By similarity, we mean your perception of one video being visually close to another one.'

For each reference video whose stimulus identity was described as stimulus A, we presented as options five different combinations: STST-A versus STPS-A, STST-A versus STST-B, STPS-A versus STPS-B, STPS-A versus STST-B and STST-A versus STPS-B. We denoted as B the STST and STPS videos having as target original videos not the one being presented as the reference. As there were three original videos, the B stimulus could be one of the two remaining ones. So, each of the combinations having the type B stimulus to be presented was shown twice with the two remaining ones. The order of appearance of the two options was randomized, as well as the trial order. We computed the preference score for the STST videos in the combination STST-A versus STPS-A as the number of times each participant selected the STST-A option divided by the number of trials of that combination. All other combinations were evaluated as the accuracy in selecting the option having the type A stimulus. Statistical comparisons were evaluated using two-tailed paired $t$-tests and Cohen's $d$ as the effect size metric.

### Stimulus generation for spatiotemporal factorization studies

We generated a metamer stimulus that was matched in space to one video and in time to another, a process that can be conceived of as spatiotemporal factorization. We used the same three videos used to generate dynamic metamer stimuli for object-recognition studies (stimulus set 1) and a new set of three videos with resolution $320 \times 180$ at 30 FPS (stimulus set 2).

To quantitatively assess the spatiotemporal similarities, we plotted the full 3D color distribution across the frames, as well as the palette, defined with the $k$-means clustering algorithm with $k = 5$, and the Kullback–Leibler (KL) divergence between each pair of stimuli in each stimulus set. We also plotted the distribution of the magnitude and angle of the optical flow across the frames, computed with the same method as above (section 'Spatiotemporal feature analysis'). We used the Euclidean distance to measure the differences between each pair of stimuli within each stimulus set for all the low-level spatial and temporal features we analyzed.

Thus, for each combination of these natural videos, we passed one video to the spatial stream $\mathcal{S}(x_t)$ and the other to the temporal stream $\mathcal{T}(x_t, x_{t-1})$. Unlike the stimulus-generation procedure above, we computed the spatial $\mathcal{L}_{st}^{\ell}$ and temporal $\mathcal{L}_{tt}^{\ell}$ texture loss components with respect to these two targets. All the hyperparameters were set the same as the previous stimulus generation procedure (section 'Stimulus generation for object-recognition studies'). To evaluate this generative procedure, we computed the Pearson correlation between the low-level features of the spatial and temporal target videos and the low-level features of the spatiotemporally factorized STST video.

### Testing spatiotemporally factorized metamers in humans and deep visual models

We performed a proof-of-principle study on the potential application of spatiotemporal factorization in both humans and deep visual models. We collected data in human participants for the 2AFC spatiotemporal perceptual similarity task, for both stimulus set 1 ($n = 13$, mean age = 30.3 (3.8 s.d.), four females) and stimulus set 2 ($n = 12$, mean age = 29.3 (3.7 s.d.), six females). Among both data collections, five participants completed the task after the previous video-captioning task and the 2AFC perceptual similarity task. As in the previous 2AFC task, each trial consisted of three videos (the reference and the remaining options), interleaved by 300 ms and presented with a 15° visual angle. The stimuli were the natural videos from each stimulus set and all the combinations from mixing them for the spatial or temporal stream, with a total of 15 stimuli. Participants were instructed to report which of the two option videos (either the spatial or temporal original target video) was spatially or temporally more similar to the reference video (one of the mixed combinations), by pressing the D and L keys to select the first or second options, respectively. At the beginning of each trial, a visual cue signaled whether similarity was to be judged in space or time. For the trials in which the reference was a video with only one target video (no mixing), we showed as options the spatial-temporal target and another natural video from the stimulus set. Because there were two alternate natural videos, we repeated this type of trial twice. To provide a consistent definition of video similarity, we gave the same instruction as for the previous 2AFC task. For spatial trials, we asked to focus on the shapes, patterns, colors and spatial arrangements in the videos. For the temporal trials, we asked to focus on the motion, timing, rhythm and sequence of the events in the videos. Subjects were also explicitly instructed to ignore temporal aspects during the spatial trials and vice versa. Behavior was quantified as the accuracy of the participants in selecting the target option corresponding to the cued feature dimension. We compared accuracies against chance level (0.5) and between spatial and temporal accuracies across participants using two-tailed paired $t$-tests and Cohen's $d$ as the effect size metric.

To test our spatiotemporally factorized videos on deep vision models trained for video classification in a comparable way to humans, we used the same ResNet18-3D and ResNet18-MC3 models used in the previous analysis (section 'Testing vision models on metamer stimuli'). We extracted early and late layer activations, as defined above, from each combination of mixing video as well as from their spatial and temporal targets. We included in the analyses only the mixed video combinations (off-diagonal), as the unmixed (on-diagonal) videos corresponded to the STST videos generated for object recognition above. We computed the CKA score between the mixed video and its respective spatial and temporal targets for each model, layer stage and stimulus set.

### Reporting summary

Further information on research design is available in the Nature Portfolio Reporting Summary linked to this Article.

### Data availability

The Kinetics400 dataset is available at https://github.com/cvdfoundation/kinetics-dataset. Generated video data are available as Supplementary Videos 1–3. Behavioral data for the human experiments are available as Supplementary Data 1. Source data are provided with this paper.

### Code availability

Code for our algorithm is available at https://github.com/antonigreco/STST (ref. 92).

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

## Acknowledgements

This study was supported by the European Research Council (ERC; https://erc.europa.eu/, CoG 864491 to M.S.) and by the German Research Foundation (DFG; https://www.dfg.de/, projects 276693517 (SFB 1233; to M.S.) and SI 1332/6-1 (SPP 2041; to M.S.)).

## Author contributions

A.G. provided conceptualization, software, methodology, investigation, formal analysis, visualization, writing of the original draft, and review and editing. M.S. provided conceptualization, supervision, resources, project administration, funding acquisition, and review and editing of the paper.

## Competing interests

The authors declare no competing interests.

## Additional information

**Correspondence and requests for materials** should be addressed to Antonino Greco or Markus Siegel.

# Reporting Summary

## Statistics

For all statistical analyses, confirm that the following items are present in the figure legend, table legend, main text, or Methods section.

| n/a | Confirmed | |
|---|---|---|
| ☐ | ☒ | The exact sample size (*n*) for each experimental group/condition, given as a discrete number and unit of measurement |
| ☐ | ☒ | A statement on whether measurements were taken from distinct samples or whether the same sample was measured repeatedly |
| ☐ | ☒ | The statistical test(s) used AND whether they are one- or two-sided *Only common tests should be described solely by name; describe more complex techniques in the Methods section.* |
| ☒ | ☐ | A description of all covariates tested |
| ☐ | ☒ | A description of any assumptions or corrections, such as tests of normality and adjustment for multiple comparisons |
| ☐ | ☒ | A full description of the statistical parameters including central tendency (e.g. means) or other basic estimates (e.g. regression coefficient) AND variation (e.g. standard deviation) or associated estimates of uncertainty (e.g. confidence intervals) |
| ☐ | ☒ | For null hypothesis testing, the test statistic (e.g. *F*, *t*, *r*) with confidence intervals, effect sizes, degrees of freedom and *P* value noted *Give P values as exact values whenever suitable.* |
| ☒ | ☐ | For Bayesian analysis, information on the choice of priors and Markov chain Monte Carlo settings |
| ☒ | ☐ | For hierarchical and complex designs, identification of the appropriate level for tests and full reporting of outcomes |
| ☐ | ☒ | Estimates of effect sizes (e.g. Cohen's *d*, Pearson's *r*), indicating how they were calculated |

*Our web collection on statistics for biologists contains articles on many of the points above.*

## Software and code

Policy information about availability of computer code

| Data collection | Behavioural data were collected using the Psychopy software v2024.1.5 |
|---|---|
| Data analysis | Data analyses has been carried out using Python 3.9, TensorFlow 2.4  and Pytorch 2.2. We also used the Sentence Transformers library (version 3.0.1) for the analysis of the video caption data. The code for reproducing the STST algorithm is available at github.com/antoninogreco/STST |

For manuscripts utilizing custom algorithms or software that are central to the research but not yet described in published literature, software must be made available to editors and reviewers. We strongly encourage code deposition in a community repository (e.g. GitHub). See the Nature Portfolio guidelines for submitting code & software for further information.

## Data

Policy information about availability of data

All manuscripts must include a data availability statement. This statement should provide the following information, where applicable:
- Accession codes, unique identifiers, or web links for publicly available datasets
- A description of any restrictions on data availability
- For clinical datasets or third party data, please ensure that the statement adheres to our policy

The Kinetics400 dataset is available at github.com/cvdfoundation/kinetics-dataset. Generated video data are available as Supplementary Video 1-3 with this

manuscript. and bBehavioral data of the human experiments are available as Ssupplementary materialData 1 with this manuscript. Source data for Figures 2-6 are provided with this paper.

# Human research participants

Policy information about studies involving human research participants and Sex and Gender in Research.

| | |
|---|---|
| Reporting on sex and gender | - video captioning task, 8 males and 6 females<br>- 2AFC perceptual similarity task, 7 males and 6 females<br>- 2AFC spatiotemporal perceptual similarity task, stimulus set 1 9 males and 4 females, and stimulus set 2 6 males and 6 females |
| Population characteristics | - video captioning task N = 14, mean age = 30.4 (3.1 SD)<br>- 2AFC perceptual similarity task N = 13, mean age = 30.0 (2.8 SD)<br>- 2AFC spatiotemporal perceptual similarity task, stimulus set 1 N = 13, mean age = 30.3 (3.8 SD), and stimulus set 2 N = 12, mean age = 29.3 (3.7 SD) |
| Recruitment | Participants were recruited via mailing lists at the University of Tübingen, Germany, as well as from the local community. Some participants were familiar with neuroscience or psychology in general, but not with the specific hypotheses of the study. We do not expect a self-selection bias or other bias to significantly impact our results because the task relied on fundamental cognitive processes that do not require specific expertise. |
| Ethics oversight | All the experiments were conducted in accordance with the Declaration of Helsinki and approved by the ethics committee of the University of Tübingen. All subjects gave informed consent. |

Note that full information on the approval of the study protocol must also be provided in the manuscript.

# Field-specific reporting

Please select the one below that is the best fit for your research. If you are not sure, read the appropriate sections before making your selection.

☐ Life sciences    ☒ Behavioural & social sciences    ☐ Ecological, evolutionary & environmental sciences

For a reference copy of the document with all sections, see nature.com/documents/nr-reporting-summary-flat.pdf

# Behavioural & social sciences study design

All studies must disclose on these points even when the disclosure is negative.

| | |
|---|---|
| Study description | Quantitative study investigating human judgments on model metamers for object recognition and spatiotemporally factorized stimuli |
| Research sample | Sample size was chosen in accordance with the typical sample sizes used in similar studies within the field, ensuring consistency and comparability with established research practices. Participants were recruited via mailing lists at the University of Tübingen, Germany, as well as from the local community. Some participants were familiar with neuroscience or psychology in general, but not with the specific hypotheses of the study. For the three tasks, the sample statistics were the following:<br>- video captioning task N = 14, mean age = 30.4 (3.1 SD)<br>- 2AFC perceptual similarity task N = 13, mean age = 30.0 (2.8 SD)<br>- 2AFC spatiotemporal perceptual similarity task, stimulus set 1 N = 13, mean age = 30.3 (3.8 SD), and stimulus set 2 N = 12, mean age = 29.3 (3.7 SD) |
| Sampling strategy | We randomly sampled participants from the local community with the only constrain of having no history of psychiatric or neurological disorder and a normal or corrected to normal vision. Sample size was chosen in accordance with the typical sample sizes used in similar studies within the field, ensuring consistency and comparability with established research practices. |
| Data collection | We used Psychopy software to perform the experiments. Participants responded either with some keys on the keyboard or using all the keys for textual description. Only the researcher and the volunteer were present during the experiment and the researcher was blind to the assignment of the experimental conditions. |
| Timing | The data collection took place from july 2024 to august 2024 |
| Data exclusions | No data were excluded |
| Non-participation | No participant dropped out |
| Randomization | We did not use any randomization as there were no experimental groups. |

# Reporting for specific materials, systems and methods

We require information from authors about some types of materials, experimental systems and methods used in many studies. Here, indicate whether each material, system or method listed is relevant to your study. If you are not sure if a list item applies to your research, read the appropriate section before selecting a response.

## Materials & experimental systems

| n/a | Involved in the study |
|-----|-----------------------|
| ☒ ☐ | Antibodies |
| ☒ ☐ | Eukaryotic cell lines |
| ☒ ☐ | Palaeontology and archaeology |
| ☒ ☐ | Animals and other organisms |
| ☒ ☐ | Clinical data |
| ☒ ☐ | Dual use research of concern |

## Methods

| n/a | Involved in the study |
|-----|-----------------------|
| ☒ ☐ | ChIP-seq |
| ☒ ☐ | Flow cytometry |
| ☒ ☐ | MRI-based neuroimaging |

nature portfolio | reporting summary

March 2021

