## [Peer Review File · Nature Computational Science]

A spatiotemporal style transfer algorithm for dynamic visual stimulus generation

Corresponding Author: Dr Antonino Greco

Version 0:

Decision Letter:

** Please ensure you delete the link to your author homepage in this e-mail if you wish to forward it to your co-authors. **

Dear Dr Greco,

Your manuscript "A spatiotemporal style transfer algorithm for dynamic visual stimulus generation" has now been seen by 3 referees, whose comments are appended below. You will see that while they find your work of interest, they have raised points that need to be addressed before we can make a decision on publication, as also described in our earlier communication.

The referees' reports seem to be quite clear. Naturally, we will need you to address **all** of the points raised.

While we ask you to address all of the points raised, the following points need to be substantially worked on:

- Better compare against related work on video style transfer and neural style transfer. This should come not only in form of discussions, but also with experiments to quantify the improvement when possible. If experiments cannot be performed to compare against related work, please specify why.
- Better discuss and justify the methodological novelty of the proposed method.
- Increase the size of the dataset used for evaluation.
- Include an ablation study to the evaluation section.
- Add a demonstration (with a real-world, practical application) of the usefulness of the proposed algorithm.

Please use the following link to submit your revised manuscript and a point-by-point response to the referees' comments (which should be in a separate document to any cover letter):

Link Redacted

** This url links to your confidential homepage and associated information about manuscripts you may have submitted or be reviewing for us. If you wish to forward this e-mail to co-authors, please delete this link to your homepage first. **

To aid in the review process, we would appreciate it if you could also provide a copy of your manuscript files that indicates your revisions by making use of Track Changes or similar mark-up tools. Please also ensure that all correspondence is marked with your Nature Computational Science reference number in the subject line.

In addition, please make sure to upload a Word Document or LaTeX version of your text, to assist us in the editorial stage.

To improve transparency in authorship, we request that all authors identified as 'corresponding author' on published papers create and link their Open Researcher and Contributor Identifier (ORCID) with their account on the Manuscript Tracking System (MTS), prior to acceptance. ORCID helps the scientific community achieve unambiguous attribution of all scholarly contributions. You can create and link your ORCID from the home page of the MTS by clicking on 'Modify my Springer Nature account'. For more information please visit www.springernature.com/orcid.

We hope to receive your revised paper within three weeks. If you cannot send it within this time, please let us know.

Best,
Fernando

--

Fernando Chirigati, PhD
Chief Editor, Nature Computational Science
Nature Portfolio

Reviewers comments:

Reviewer #1 (Remarks to the Author):

This work 1) introduces a method for synthesizing videos potentially useful for vision research, 2) characterizes the synthetic videos in terms of low-level statistics and deep-net representations, and 3) showcases using the videos to study PredNet, a computational model relevant to vision science. Overall, I think the paper makes useful contributions and is well-written, although some results need more thoroughness to better support the claims, and the synthesis method suggests a proximal variant that is untested but relates to much existing work.

Main comments

1. The novel synthesis method is named style transfer, but the paper only demonstrates an application on metamer synthesis (i.e., texture matching). Indeed, the paper Methods (line 389) indicate that only the texture loss is enabled, while the content loss is excluded. Given this, I find the method's name misleadingly broad and suggest that testing a setting more analogous to style transfer, with the content loss terms enabled, may significantly increase the study's relevance, with an (arguably) reasonable amount of extra effort. At the same time, how does this study relate to the literature on video style transfer? E.g., Ruder et al., IJCV 2018, Huang et al., CVPR 2017, Gao et al., WACV 2020. A case might be made that, unlike these prior methods, the current study focuses on matching level statistics, making the videos more useful to some vision research questions. However the authors choose to position this study, they should discuss these highly relevant previous studies.
2. All of the paper results depend on three example videos. While three examples suffice for illustration purposes, more videos should be tested to establish the results more conclusively. Testing more videos is crucial for analyses that show variations across videos or frames. For example, STPS seems to better match the original video than STST in Fig. 2b—e.g., the contrast in video 2, around frames 0–20, and the pixel change in video 3, around frames 10–60. The results from testing many videos should be summarized appropriately, showing both general trends and variations across videos and frames. For example, in Fig. 4c, the frame-to-frame similarity (for PredNet predictions from STST and original videos) has a small difference when averaged across all frames, but this small difference is probably statistically significant across frames. Also in Fig. 4c, the direction of difference (STST vs. original) is opposite for SSIM and cSSIM, and video 2 is an outlier for cSSIM, showing STST > original, opposite to videos 1 and 3. Thus, more videos need to be tested to establish a consistent trend.
3. While the results show STST videos are better matched to the original videos than STPS videos in many statistics, are the results on PredNet (Fig. 4) already obtainable using STPS videos? Comparing STST and STPS videos in this example setting is necessary to show that STST videos may be more useful for vision research.

Minor comments

- It would be informative to view and compare the original, STST, and STPS videos for the three examples; the videos can be shared as review material.
- Please provide an estimate of the run time of the synthesis method.
- Please verify, in eqn. (2), whether the factor $1/N_l$ is extraneous.
- On lines 204–205, please consider using a different symbol to represent the octave because the letter o is easily confused with the number 0.
- In Fig. 3, it would be more informative to show how the CKA score varies as a function of layer depth across more layers. Such an analysis would characterize how the metamer gradually diverges from the original for higher-level representations. The current presentation of the CKA as a function of frame number has no intrinsic meaning because the reader cannot see why different frames lead to different results. For completeness, it will also be nice to compare STPS to STST in this figure.
- Lines 305–310. The hypothesis ('we expected [...]') and the next two sentences ('In contrast [...] structure.') are confusing because the hypothesis seems to be about comparing natural and metameric videos but the sentences compare PredNet layers within a video. I also don't understand why we should expect the error size to be smaller (or larger) progressively along PredNet layers. If the main comparison is between video types, Fig. 4d should be organized differently—as 2 lines per subplot, 4 subplots—to directly compare metamer and original videos.
- Lines 309–310, '[T]he amount of prediction error across levels did not show a clear sorted pattern [...].' This sentence is ambiguous because there is a clear sorted pattern: the error size in layer $2 > 3 > 4 > 1$. It is only that the sorting does not follow the layer depth.

Reviewer #2 (Remarks to the Author):

The paper describes an algorithm (called STST) that generates spatiotemporal visual stimuli (i.e. videos) that may be useful to probe biological and artificial vision systems. The algorithm is based on previous work that used artificial neural networks to learn and segregate the style of an image or painting from its content, and then used that learned segregation to generate new images that combine the content from one source image with the style from another. The current algorithm extends this method of segregating style (i.e. low-level representations in a pretrained vision DNN) from content (i.e. high-level representations) to the temporal domain, creating a spatiotemporal image generation framework that is able to generate image frame sequences that match the neural network representations at various levels of the representational hierarchy as specified by hyperparameters of a combined objective function; the authors refer to these generated sequences as model metamers. The authors propose that this generative algorithm is a versatile tool for vision research.

A - Technically, the work seems sound, building on previous methods and representational modules of spatial and temporal visual information as well as optimization constraints and methods necessary for the generative process. Conceptually, however, the paper lacks a clear demonstration of the promised additional value this generative stimulus framework provides to the vision science community: what does the method allow us to do that we cannot do at the moment? what potential new insights into vision will it help to uncover?

The authors present two tests for their framework to demonstrate its function and advantages. First they show that while performing equally well at the spatial feature level, the new framework provides stimuli that better match the low-level temporal characteristics of natural video scenes (i.e. optic flow) compared to an alternative generative algorithm (STPS). This does not come as a surprise given that the temporal stream of their model is based on a pretrained CNN optimized for predicting optical flow patterns, while the alternative model had a much simpler procedure to maintain some of the basic temporal structure of the video sequences (i.e. phase scrambling); thus this seems not a particularly fair comparison with a predictable outcome.

The second test consisted of showing that the generated frames lead to very similar activation patterns in early but not late layers of image classification models such as ResNet50 when compared to the activation patterns of their natural stimulus counterparts (Fig.3). Again, this comes not as a surprise since the framework was designed to match those activation patterns ("model metamers") in such deep convolutional image classification networks (VGG19) in the first place. It is a control to show that the framework does what it is supposed to do.

One application is presented, testing whether PredNet, a recurrent neural network trained to predict the next frame of an video sequence, operates on high-level representations or not. Besides the fact that questions like these are notoriously difficult to answer due to all the different constraints and boundary conditions involved in building such ANN systems (i.e. specific training data, task, performance metrics etc.), the missing aspect here is to not show that already existing, alternative methods (e.g. STPS) are insufficient to address this question.

The work and the description in the paper simply do not provide a convincing demonstration of the usefulness of the proposed algorithm and its advantage for the field.

B - Other conceptual aspects of the work may also require a more critical analysis and evaluation. Previous methods such as the Portilla/Simoncelli texture model aimed to generate (texture) frames that are precisely matched to their natural counterparts in terms of their low-level statistics of simple, well-specified feature kernels. This allows for a clear interpretations of what aspects of the stimulus are matched. In contrast, the approach here relies on matching layer representations of a specific pretrained convolutional DNN, creating stimuli that the authors refer to as "model metamers". These metamers are naturally dependent on the precise choice of network (VGG19) and its training, and thus ultimately on the specific feature space represented - in that sense the method uses quite subjective metrics. The open question that is not addressed with the work is how the generative process depends on these choices.

The inclusion of a temporal representation stream seems useful. However, the particular choice of the network limits temporal structures to the extent of only adjacent frames in the video sequence. This seems sufficient to approximately compute optic flow, but cannot capture temporal structures at larger timescales. There are likely practical aspects that makes considering longer frame sequences more difficult, but the particular choice and its limitations should be evaluated and discussed.

C - Finally, there are some technical aspects of the method that the paper does not explain/address, yet which seem important for the behavioral as well as a practical application of the algorithm:

Hyperparameters (alpha, beta ...): these parameters allow the algorithm to generate arbitrary metamers that match their natural counterparts to various degrees at various representational levels. This degree of freedom of the algorithm is not explored in the paper. One would expect a simple demonstration that the algorithm can successfully generate such metamers with various degree of shifts from low-to-high level matching.

One concern is that because the algorithm seems to need substantial preconditioning to create proper results, certain range settings of the hyperparameters may actually not be practically possible as they would not lead to stable or desirable solutions.

Another open choice is the selection of the layers in the overall matching objective (Eqs.4,5,6). How does the generative process depend on the particular choices? The author list which layer they chose for their specific examples, but do not evaluate or discuss the rationale for their choices.

In summary, the presentation of the propose generative framework seems a bit premature with many open questions not being addressed. This includes a demonstration of the framework's scientific value.

References:

Portilla/Simoncelli (2000), A Parametric Texture Model Based on Joint Statistics of Complex Wavelet Coefficients, International Journal of Computer Vision 40(1), 49–71

Reviewer #3 (Remarks to the Author):

Summary

The paper presents a Spatiotemporal Style Transfer (STST) algorithm for generating dynamic visual stimuli in vision research. The STST algorithm is based on a two-stream deep neural network model that separates spatial and temporal features to synthesize video stimuli. This approach allows researchers to create "model metamers," dynamic stimuli that match the low-level spatiotemporal features of natural videos but lack high-level semantic content. The study demonstrates that the generated stimuli can be used to investigate object recognition and probe internal representations in predictive coding networks.

Novelty

The STST algorithm represents a significant advancement in the field of dynamic visual stimulus generation, filling a gap left by previous methods that primarily focused on static images. By integrating both spatial and temporal aspects, this work offers a novel approach to potentially further understanding visual information processing in both artificial and biological systems.

However, the algorithm employed to generate the stimuli itself seems to be a straightforward extension of neural style transfer to videos, lacking sufficient novelty in methodology and appropriate discussion of prior work. Meanwhile, although the claimed objective of the paper appears novel, it does not sufficiently demonstrate an application of the proposed approach in directly studying biological systems, as all experiments were validated by neural networks rather than through subjective studies and psychophysical evaluations.

Methodology, Clarity, and Presentation

The paper is well-organized, with a logical flow from introduction to methodology, results, and discussion. The figures effectively illustrate the STST algorithm and its outputs.

In Eqn 4, the notations L_{sc} and L_{st} were never introduced. I assume they refer to "spatial content" and "spatial texture" respectively?

The total variation loss introduced by Eqn 9 does not seem to preserve edges, as large variation over edges can also be penalized by Eqn 9.

The result images presented in the paper are too small to understand the content effectively. The supplementary materials also did not include any videos processed by the authors for evaluation. Only the source video of the animal scene has been included.

Evaluation

The number of videos used for evaluation and validation is too small and subject to cherry-picking. The authors are advised to select a public dataset without any biased manual selection of content for a fair and extensive evaluation. Meanwhile, the hypotheses proposed by the authors have only been validated by neural networks. Since the paper aims to apply the proposed generated stimuli to test specific hypotheses in both biological and artificial systems, I was wondering if there are any hypotheses that can be validated through rigorous psychophysical experiments. Finally, an ablation study for the total variation loss and the color matching method is highly recommended but currently missing.

Reviewer #3 (Remarks on code availability):

I ran into the following error when attempting to run "python synthesize.py -gpu 1 -tar Animal.mp4"

Traceback (most recent call last):

File "STST/synthesize.py", line 115, in <module>

gen = stst.generate(config)

^^^^^^^^^^^^^^^^^^^^^^^^^^^^^^^^

File "STST/SpaceTimeStyleTransfer.py", line 384, in generate

log = open(self.logname,'w')

^^^^^^^^^^^^^^^^^^^^^^^^^^^^^^^^

FileNotFoundError: [Errno 2] No such file or directory: 'log/log_240607_1856_.txt'

Version 1:

Decision Letter:

Our ref: NATCOMPUTSCI-24-0826A

23rd October 2024

Dear Dr. Greco,

Thank you for submitting your revised manuscript "A spatiotemporal style transfer algorithm for dynamic visual stimulus generation" (NATCOMPUTSCI-24-0826A). It has now been seen by the original referees and their comments are below. The reviewers find that the paper has improved in revision, and therefore we'll be happy in principle to publish it in Nature Computational Science, pending minor revisions to satisfy the referees' final requests and to comply with our editorial and formatting guidelines.

TRANSPARENT PEER REVIEW

Nature Computational Science offers a transparent peer review option for original research manuscripts. We encourage increased transparency in peer review by publishing the reviewer comments, author rebuttal letters and editorial decision letters if the authors agree. Such peer review material is made available as a supplementary peer review file. **Please remember to choose, using the manuscript system, whether or not you want to participate in transparent peer review.**

Thank you again for your interest in Nature Computational Science. Please do not hesitate to contact me if you have any questions.

Sincerely,
Fernando

--

Fernando Chirigati, PhD
Chief Editor, Nature Computational Science
Nature Portfolio

ORCID

Reviewer #1 (Remarks to the Author):

I appreciate the additional analyses and experiments in this revision, which has addressed my previous comments. Furthermore, the psychophysics results are instructive, and the experiment on factorizing spatiotemporal information is very nice. Below are some comments on the new material.

1. While I appreciate the expanded analyses in Fig. 4, I do not follow the reasoning surrounding SSIM, PredNet, and high-level information. The SSIM metric is not expected to capture high-level information, so how do the SSIM results show that high-level information plays no role in PredNet? The results could be because the low-level information in STST and STPS

videos is easier (per SSIM) for PredNet to capture than the low-level information in natural videos. This does not show whether or not PredNet captures high-level information.

2. I'm unsure that the results in Fig. 7 show people are biased for spatial features in similarity judgments. The synthetic videos based on current methods may be better at mimicking spatial features than temporal features. This possibility could also explain the analogous results in models. Moreover, in the results for models, the absolute accuracy difference is small on the left side of Fig. 7e and f (< 0.06 and < 0.01 respectively).

Minor comments:

1. Regarding whether the normalization factor of $1/N_l$ is extraneous in Eqn. (2), substituting Eqn. (2) into Eqn. (3) suggests that the sum is normalized by a total factor of N^4_l , whereas the sum is only over N^2_l elements. It would be helpful for the authors to share an intuition as to why.

2. Line 220: Should the H_l and W_l be indexed by o (calligraphic o) instead of σ (sigma)? It looks like the calligraphic o is intended to be the index for the octave levels.

Reviewer #2 (Remarks to the Author):

In my assessment of the original submission, my main concern was the lack of a thorough and convincing demonstration of the value and usefulness of the proposed generative method. The authors have addressed this with the inclusion of new experimental data and a revised and extended discussion. The work has substantially improved; I have no further comments.

Reviewer #3 (Remarks to the Author):

The revised manuscript has effectively addressed most of my concerns, improving both the depth of the analysis and the clarity of the presentation. The three newly added psychophysical experiments in humans provide evidence supporting the effectiveness of the proposed Spatiotemporal Style Transfer (STST) algorithm for studying biological systems. A minor concern is that the conclusion drawn from the third experiment regarding the spatial bias appears to rely on the assumption that the metamers generated by STST are the ground truth. However, previous experiments only demonstrate that STST performs better in preserving low-level features than existing methods, rather than proving its results are exact and error-free. I hope other reviewers with expertise in psychophysics can further comment on this. In addition, the evaluation on a larger and more diverse dataset, along with the addition of the ablation study, is also appreciated. Overall, the revisions have significantly enhanced the manuscript, and I recommend acceptance after addressing these remaining points.

Version 2:

Decision Letter:

21st November 2024

Dear Dr. Greco,

I am delighted to tell you that your manuscript NATCOMPUTSCI-24-0826B has been accepted for publication in Nature Computational Science.

We will be publishing your paper on an accelerated schedule. **Please carefully review the details below and contact us immediately at computationalscience@nature.com if you have any travel plans or other conflicts that may make you unable to respond to us for the next 5-7 days.**

In approximately 2 business days you will receive a link to choose the appropriate publishing options for your paper and complete the appropriate grant of rights necessary to publish your work. As it is vital that this process not be delayed, we strongly encourage you to [whitelist](https://www.simpleminds.com/how-to-check-your-spam-filter-and-whitelist-emails/) the email address do-not-reply@springernature.com to ensure that this message is received.

You will receive a link to your electronic proof via email with a request to make any necessary corrections as soon as possible. You will find that we have made minor changes to enhance the clarity of the text and to ensure that your paper conforms to the journal's style so we ask that you review these proofs carefully to ensure that we have not inadvertently introduced errors or altered the sense of your text in any way.

Please return your proof within 24 hours of receiving it. If you have any questions about your proofs or anticipate any delays please contact rjsproduction@springernature.com immediately.

Once a publication date is set for your paper, the Springer Nature press office will be in touch with the full embargo details. We request that you do not send out your own publicity or contact any journalists until you hear from us that the paper has a confirmed publication date.

If you would like to inform your Public Relations or Press Office about your paper, we suggest that you do so immediately to

allow them as much time as possible to prepare an appropriate press release and organize publicity if they choose to do so. Please include your manuscript tracking number NATCOMPUTSCI-24-0826B and the name of the journal, which they will need if they contact our press office.

Please note that Nature Computational Science is a Transformative Journal (TJ). Authors may publish their research with us through the traditional subscription access route or make their paper immediately open access through payment of an article-processing charge (APC). Authors will not be required to make a final decision about access to their article until it has been accepted. Find out more about Transformative Journals

Authors may need to take specific actions to achieve compliance with funder and institutional open access mandates. If your research is supported by a funder that requires immediate open access (e.g. according to Plan S principles) then you should select the gold OA route, and we will direct you to the compliant route where possible. For authors selecting the subscription publication route, the journal's standard licensing terms will need to be accepted, including self-archiving policies. Those licensing terms will supersede any other terms that the author or any third party may assert apply to any version of the manuscript.

If you have any questions about our publishing options, costs, Open Access requirements, or our legal forms, please contact ASJournals@springernature.com.

An online order form for reprints of your paper is available at https://www.nature.com/reprints/author-reprints.html. All co-authors, authors' institutions and authors' funding agencies can order reprints using the form appropriate to their geographical region.

Sincerely,

--
Fernando Chirigati, PhD
Chief Editor, Nature Computational Science
Nature Portfolio

P.S. Click here if you would like to recommend Nature Computational Science to your librarian - this will link directly to the Recommend page.

<http://www.nature.com/subscriptions/recommend.html#forms>

** Visit the Springer Nature Editorial and Publishing website at www.springernature.com/editorial-and-publishing-jobs for more information about our career opportunities. If you have any questions please click here.**

We thank the reviewers and the editor for their positive assessment of our work, and for their thoughtful and constructive comments. These comments were very helpful for improving the manuscript. Following the reviewers' advice, we did not only substantially revise the entire manuscript but performed several new experiments with intriguing results that we added to the revised manuscript. We are confident that these changes and additions address all concerns of the reviewers and substantially strengthen the manuscript.

Main changes and new results:

- We performed three new psychophysical experiments in humans using artificial stimuli generated with the STST algorithm in comparison to previous methods. In the first experiment we combined a video annotation paradigm and LLMs to show how STST can be used to selectively remove semantic video content. The second psychophysical experiment provides direct perceptual evidence that STST effectively preserves low-level visual features while removing high-level content. The third experiment introduces an entirely new and unique use case of the STST algorithm, which is to generate stimuli that factorize the spatial and temporal features from two different target videos, i.e. that mix the spatial and temporal features from two different movies. To the best of our knowledge, the generation of such stimuli was previously impossible. We provide new results on both humans and deep vision models on these stimuli, that suggest a spatial bias in visual similarity judgements in both artificial and biological systems. We included all these new experiments and results in the revised manuscript.

As requested, these findings do not only show the novelty of the STST algorithm and its improvement in comparison to previous methods, but also demonstrate its usefulness for real-world, practical applications in vision science that were previously out of reach.

- As suggested, we performed ablation experiments to pinpoint the contribution of critical algorithmic components for the performance of the STST algorithm.
- As suggested, we increased the size of the dataset to 100 videos randomly sampled from the Kinetics400 dataset and validated all key results on this dataset.
- We revised the entire manuscript, added 3 new main figures, 4 new supplementary figures, 3 new supplementary videos and updated the software supplementary material.

Below, we provide point-by-point responses to all three reviewers. To ease navigation, reviewer comments are written in **black** and our replies in **blue**.

Reviewer #1:

This work 1) introduces a method for synthesizing videos potentially useful for vision research, 2) characterizes the synthetic videos in terms of low-level statistics and deep-net representations, and 3) showcases using the videos to study PredNet, a computational model relevant to vision science. Overall, I think the paper makes useful contributions and is well-written, although some results need more thoroughness to better support the claims, and the synthesis method suggests a proximal variant that is untested but relates to much existing work.

We thank the Reviewer the positive and constructive assessment of our work.

Main comments

1. The novel synthesis method is named style transfer, but the paper only demonstrates an application on metamer synthesis (i.e., texture matching). Indeed, the paper Methods (line 389) indicate that only the texture loss is enabled, while the content loss is excluded. Given this, I find the method's name misleadingly broad and suggest that testing a setting more analogous to style transfer, with the content loss terms enabled, may significantly increase the study's relevance, with an (arguably) reasonable amount of extra effort. At the same time, how does this study relate to the literature on video style transfer? E.g., Ruder et al., IJCV 2018, Huang et al., CVPR 2017, Gao et al., WACV 2020. A case might be made that, unlike these prior methods, the current study focuses on matching level statistics, making the videos more useful to some vision research questions. However the authors choose to position this study, they should discuss these highly relevant previous studies.

We thank the Reviewer for bringing up this important point. We agree with the reviewer. We followed the reviewer's suggestion and added a section in the discussion in which clarify the spectrum of possible application of the proposed algorithm even beyond the cases that were employed in the present work. As suggested by the reviewer, in this new section, we now also discuss the relation of the new STST algorithm to previous studies in the field of video style transfer. The corresponding section now reads: "The proposed framework may foster vision science beyond the applications showcased here. For instance, in the present examples we only manipulated the texture (style) of the spatiotemporal targets, either from the same natural video or by factorizing space or time with two different movies. Future applications of the STST algorithm may explore all possible combinations of content and texture loss along the spatial and temporal dimension with up to four distinct target videos. For example, another application is the so called video style transfer⁸⁷⁻⁸⁹, in which the content from one target video is combined with the style from another video or image. For this application, other algorithms have been proposed, some of which also exploit optical flow information to improve perceptual stability⁸⁸. The proposed STST algorithm differs from these video style transfer alternatives⁸⁷⁻⁸⁹ in terms of model architecture (two-stream approach), flexibility, and its ability to match spatiotemporal stimulus statistics at various levels. These aspects are especially relevant when conceiving our approach as a dynamic visual stimulus generation framework. This is in particular the case for the field of NeuroAI where the comparison between biological and artificial systems is increasingly evaluated by means of out-of-distribution stimuli^{90,39,91,92}, i.e. stimuli that diverge from the training set of artificial systems and what biological systems faces during their lifetimes. Along this line, the STST framework opens a new out-of-distribution regime for video stimuli." (line 564).

Concerning the algorithm's name, we understand the reviewer's reasoning. Yet, the proposed algorithm is a natural spatiotemporal extension of the classical neural style transfer algorithm and the term 'style transfer' has been well established for the type of algorithm and implementation at hand. Thus, we feel that the original name is very meaningful and clear for readers, and we opted to stick with the original name.

2. All of the paper results depend on three example videos. While three examples suffice for illustration purposes, more videos should be tested to establish the results more conclusively. Testing more videos is crucial for analyses that show variations across videos or frames. For example, STPS seems to better match the original video than STST in Fig. 2b—e.g., the contrast in video 2, around frames 0–20, and the pixel change in video 3, around frames 10–60. The results from testing many videos should be summarized appropriately, showing both

general trends and variations across videos and frames. For example, in Fig. 4c, the frame-to-frame similarity (for PredNet predictions from STST and original videos) has a small difference when averaged across all frames, but this small difference is probably statistically significant across frames. Also in Fig. 4c, the direction of difference (STST vs. original) is opposite for SSIM and cSSIM, and video 2 is an outlier for cSSIM, showing STST > original, opposite to videos 1 and 3. Thus, more videos need to be tested to establish a consistent trend.

We thank the Reviewer for raising this important point. We followed the reviewer's suggestion and validated all key results on a larger video set comprising 100 videos randomly sampled from the Kinetics400 dataset. This included the spatiotemporal feature analysis, deep vision model representations and PredNet performance. All our previous results were confirmed. Furthermore, we extended our results by now also adding the comparison between STST and STPS videos for the deep vision model representations and PredNet performance sections.

3. While the results show STST videos are better matched to the original videos than STPS videos in many statistics, are the results on PredNet (Fig. 4) already obtainable using STPS videos? Comparing STST and STPS videos in this example setting is necessary to show that STST videos may be more useful for vision research.

We thank the Reviewer for pointing this out. We substantially extended the analysis on the PredNet performance including STPS videos and a larger the validation dataset (100 videos from Kinetics400). Our results show how PredNet performed even better for STPS and STST videos than for natural ones further strengthening our results. Crucially, we also found a significant difference in PredNet performance for STPS and STST videos and that this is likely since STST better preserves the similarity between consecutive frames than STPS. This provides direct evidence how STST is more useful for vision research than previous methods.

Related to the usefulness of the STST method we also highlight that, in the revised manuscript, we added results from the new psychophysical experiments that show how STST performed better than previous methods (STPS). In the third of these experiments, we introduce an entirely new and unique use case of the STST algorithm, which is to generate stimuli that factorize the spatial and temporal features from two different target videos, i.e. that mix the spatial and temporal features from two different movies – an application that, to the best of our knowledge, has been previously impossible.

Minor comments

- It would be informative to view and compare the original, STST, and STPS videos for the three examples; the videos can be shared as review material.

We thank the reviewer for pointing out this. We followed the reviewer's suggestion and included the videos as supplementary material.

- Please provide an estimate of the run time of the synthesis method.

We thank the reviewer for pointing out this. We added this information to the methods section (line 694).

- Please verify, in eqn. (2), whether the factor $1/N_f$ is extraneous.

We checked this point in eq. 2 and it is correct. The Gram matrix must be normalized by the number of filters (N_f) along which it has been computed.

- On lines 204–205, please consider using a different symbol to represent the octave because the letter o is easily confused with the number 0.

We changed the notation for the octave reference to the symbol α .

- In Fig. 3, it would be more informative to show how the CKA score varies as a function of layer depth across more layers. Such an analysis would characterize how the metamer gradually diverges from the original for higher-level representations. The current presentation of the CKA as a function of frame number has no intrinsic meaning because the reader cannot see why different frames lead to different results. For completeness, it will also be nice to compare STPS to STST in this figure.

We thank the reviewer for these suggestions. We followed the reviewer and added a supplementary figure (Fig. S3) to show the comparison between STST and STPS on the validation stimulus set. Concerning the suggestion to show the CKA across all model stages, we felt that it would not add evidence to the key claim that first and last layers are differently perturbed by the lack of high-level content. Thus, considering the many new results added in the revised manuscript and in the spirit of keeping the paper as concise as possible, we opted not to add more intermediate layers.

- Lines 305–310. The hypothesis ('we expected [...]') and the next two sentences ('In contrast [...] structure.') are confusing because the hypothesis seems to be about comparing natural and metameric videos but the sentences compare PredNet layers within a video. I also don't understand why we should expect the error size to be smaller (or larger) progressively along PredNet layers. If the main comparison is between video types, Fig. 4d should be organized differently—as 2 lines per subplot, 4 subplots—to directly compare metamer and original videos.

See reply to next point.

- Lines 309–310, '[T]he amount of prediction error across levels did not show a clear sorted pattern [...].' This sentence is ambiguous because there is a clear sorted pattern: the error size in layer $2 > 3 > 4 > 1$. It is only that the sorting does not follow the layer depth.

We thank the reviewer for bringing up these points related to the prediction error analysis. We agree with the reviewer that the previous analysis was not sufficiently clear in terms of hypothesis and description. Moreover, this analysis did not add to the key results concerning the predictive performance of PredNet. As we substantially expanded this performance analysis, added many other results in the revision, and to keep the paper concise, we decided to drop this analysis in the revised manuscript.

Reviewer #2:

The paper describes an algorithm (called STST) that generates spatiotemporal visual stimuli (i.e. videos) that may be useful to probe biological and artificial vision systems. The algorithm is based on previous work that used artificial neural networks to learn and segregate the style of an image or painting from its content, and then used that learned segregation to generate new images that combine the content from one source image with the style from another. The current algorithm extends this method of segregating style (i.e. low-level representations in a pretrained vision DNN) from content (i.e. high-level representations) to the temporal domain, creating a spatiotemporal image generation framework that is able to generate image frame

sequences that match the neural network representations at various levels of the representational hierarchy as specified by hyperparameters of a combined objective function; the authors refer to these generated sequences as model metamers. The authors propose that this generative algorithm is a versatile tool for vision research.

A - Technically, the work seems sound, building on previous methods and representational modules of spatial and temporal visual information as well as optimization constraints and methods necessary for the generative process. Conceptually, however, the paper lacks a clear demonstration of the promised additional value this generative stimulus framework provides to the vision science community: what does the method allow us to do that we cannot do at the moment? what potential new insights into vision will it help to uncover?

We thank the reviewer for the constructive and positive assessment of our work. In the revised manuscript, we made substantial changes and added new experiments and results to demonstrate the added value of the proposed STST method. The changes and additions are along three lines:

First, we performed three new psychophysical experiments in humans using artificial stimuli generated with the STST algorithm in comparison to previous methods. In the first experiment we combined a video annotation paradigm and LLMs to show how STST can be used to selectively remove semantic video content. The second psychophysical experiment provides direct perceptual evidence that STST effectively preserves low-level visual features while removing high-level content. The third experiment introduces an entirely new and unique use case of the STST algorithm, which is to generate stimuli that factorize the spatial and temporal features from two different target videos, i.e. that mix the spatial and temporal features from two different movies. To the best of our knowledge, generation of such stimuli was previously impossible. We provide new results on both humans and deep vision models on these stimuli, that suggest a spatial bias in visual similarity judgements in both artificial and biological systems. We included all these new experiments and results in the revised manuscript. As requested, these findings do not only show the novelty of the STST algorithm and its improvement in comparison to previous methods, but also demonstrate its usefulness for real-world, practical applications in vision science that were previously out of reach.

Second, we performed and added results of new analyses of previous methods (STPS) in comparison to the proposed STST algorithm. Specifically, we added experiments with STPS stimuli to the analysis of PredNet and added new analysis of STPS stimuli with a large validation dataset (100 videos). These new analyses and results show how STST is better suited than STPS for the applications at hand as it better preserves critical stimulus statistics.

Third, we thoroughly revised the entire manuscript highlighting more clearly how the proposed algorithm deviates from and outperforms previous methods as well as how it can be a valuable tool for the vision science community by generating previously impossible stimuli (see discussion).

We feel that with its substantial revision and the many added experiments and results the manuscript now provides a convincing demonstration of the added value of our algorithm and its usefulness for vision science.

The authors present two tests for their framework to demonstrate its function and advantages. First they show that while performing equally well at the spatial feature level, the new framework provides stimuli that better match the low-level temporal characteristics of natural video scenes (i.e. optic flow) compared to an alternative generative algorithm (STPS). This

does not come as a surprise given that the temporal stream of their model is based on a pretrained CNN optimized for predicting optical flow patterns, while the alternative model had a much simpler procedure to maintain some of the basic temporal structure of the video sequences (i.e. phase scrambling); thus this seems not a particularly fair comparison with a predictable outcome.

We thank the reviewer for pointing this out. We agree with the reviewer. Indeed, our claim in this section was not that our results were surprising, but we aimed to confirm our expectation that the proposed algorithm outperforms the only available alternative algorithm on this purpose. We clarified this intention in the revised manuscript.

The second test consisted of showing that the generated frames lead to very similar activation patterns in early but not late layers of image classification models such as ResNet50 when compared to the activation patterns of their natural stimulus counterparts (Fig.3). Again, this comes not as a surprise since the framework was designed to match those activation patterns ("model metamers") in such deep convolutional image classification networks (VGG19) in the first place. It is a control to show that the framework does what it is supposed to do.

We agree with the reviewer. As for the previous point and in accordance with the reviewer, our aim was to confirm our expectation about the effect of STST metamer stimuli on deep vision models. Importantly, we employed different models with respect to the ones we used to generate these stimuli. Thus, although expected, our results are not completely uninformative. In particular, we believe that our findings that image models are also affected by modifications of the optical flow, as the comparison with the STPS algorithm in the new validation analyses shows, is nontrivial.

Moreover, in the revised manuscript we added results from new psychophysical experiments, in which we validated our stimuli on human observers. These experiments confirmed the results in artificial systems concerning selective feature manipulations and revealed intriguing parallels between humans and the tested artificial systems.

One application is presented, testing whether PredNet, a recurrent neural network trained to predict the next frame of an video sequence, operates on high-level representations or not. Besides the fact that questions like these are notoriously difficult to answer due to all the different constraints and boundary conditions involved in building such ANN systems (i.e. specific training data, task, performance metrics etc.), the missing aspect here is to not show that already existing, alternative methods (e.g. STPS) are insufficient to address this question.

We thank the reviewer for pointing this out. We agree with the reviewer. We substantially extended the analysis on the PredNet performance including STPS videos and a larger validation dataset (100 videos from Kinetics400). Our results show how PredNet performed even better for STPS and STST videos than for natural ones further strengthening our results. Crucially, we also found a significant difference in PredNet performance for STPS and STST videos and that this is likely since STST better preserves the similarity between consecutive frames than STPS. This provides direct evidence how STST is more useful for vision research than previous methods.

Furthermore, we would like to clarify that our claim was not about the general class of PredNet models but about the specific model we considered. Recently, this specific model attracted a lot of attention in both the machine learning and neuroscience community (see Lotter et al., ICLR 2017 and Lotter et al., Nature Machine Intelligence 2020). Our contribution is to use STST stimuli for testing the hypothesis raised by recent studies (Rane et al., ICMR 2020), that

this PredNet model is not sensitive to object-level information. Indeed, our results provide no evidence to reject his hypothesis.

The work and the description in the paper simply do not provide a convincing demonstration of the usefulness of the proposed algorithm and its advantage for the field.

As detailed above, we feel that with its substantial revision and the many added experiments and results our manuscript now provides a convincing demonstration of the added value of our algorithm and its usefulness for vision science.

B - Other conceptual aspects of the work may also require a more critical analysis and evaluation. Previous methods such as the Portilla/Simoncelli texture model aimed to generate (texture) frames that are precisely matched to their natural counterparts in terms of their low-level statistics of simple, well-specified feature kernels. This allows for a clear interpretation of what aspects of the stimulus are matched. In contrast, the approach here relies on matching layer representations of a specific pretrained convolutional DNN, creating stimuli that the authors refer to as "model metamers". These metamers are naturally dependent on the precise choice of network (VGG19) and its training, and thus ultimately on the specific feature space represented - in that sense the method uses quite subjective metrics. The open question that is not addressed with the work is how the generative process depends on these choices.

We thank the reviewer for bringing up this important point. We agree with the reviewer that DNN-based texture models are less interpretable as compared to parametric models such as the Portilla & Simoncelli model. Yet, there is substantial evidence that DNNs very well capture natural input statistics. Thus, as DNNs do not assume fixed filter responses but learn from a substantial amount of data, trained DNNs may even be better suited as generative models of natural input statistics than theoretical-driven models (see for example Walker et al., 2019 Nature Neuroscience).

Furthermore, and in accordance with the reviewer's point, we see the presented algorithm as a general framework for which the specific spatial and temporal stream models are subject to further exploration and change. Thus, the focus of the present study, which we consider already quite extensive including many tests in humans and artificial vision systems, was to introduce this new algorithm and to demonstrate its usefulness with a specific and robust implementation. This implementation is based on previous studies on image style transfer and dynamic texture synthesis. An exploration of how the algorithm may be adapted by incorporating other spatiotemporal models is beyond the scope of this study.

Along this line, we followed the reviewer's suggestion and added a section in the revised manuscript in which we discuss this point and encourage the exploration of further spatiotemporal models (line 581).

The inclusion of a temporal representation stream seems useful. However, the particular choice of the network limits temporal structures to the extent of only adjacent frames in the video sequence. This seems sufficient to approximately compute optic flow, but cannot capture temporal structures at larger timescales. There are likely practical aspects that makes considering longer frame sequences more difficult, but the particular choice and its limitations should be evaluated and discussed.

We thank the reviewer for pointing this out. We followed the reviewer's suggestion and added a section in the discussion that promotes the exploration of temporal stream models that allow to consider longer temporal scales (line 585).

C - Finally, there are some technical aspects of the method that the paper does not explain/address, yet which seem important for the behavioral as well as a practical application of the algorithm:

Hyperparameters (alpha, beta ...): these parameters allow the algorithm to generate arbitrary metamers that match their natural counterparts to various degrees at various representational levels. This degree of freedom of the algorithm is not explored in the paper. One would expect a simple demonstration that the algorithm can successfully generate such metamers with various degree of shifts from low-to-high level matching.

We thank the reviewer for pointing this out. We added a section in the discussion that not only promotes the exploration of different settings of our framework but also highlights the described settings as an almost out-of-the-box solution for many applications, as we demonstrate through the many showcased examples (from line 581).

One concern is that because the algorithm seems to need substantial preconditioning to create proper results, certain range settings of the hyperparameters may actually not be practically possible as they would not lead to stable or desirable solutions.

We agree with the reviewer that not all areas of the hyperparameter space will produce desirable stimuli. At the same time, we did not aim to propose an algorithm that works for every combination of hyperparameters. This is why we provide a detailed description of every aspect of the hyperparameters involved, of their theoretical effect, and why we encouraged to use the settings we described for the applications we showcased.

Nevertheless, we also performed new ablation experiments to characterize the influence of some hyperparameters such as the color matching and the total variation loss. We included these new results in the revised manuscript (line 292, Fig. S2).

Another open choice is the selection of the layers in the overall matching objective (Eqs.4,5,6). How does the generative process depend on the particular choices? The author list which layer they chose for their specific examples, but do not evaluate or discuss the rationale for their choices.

We thank the reviewer for pointing this out. We opted for the chosen layers since they are the current de facto standard in the style transfer and dynamic texture synthesis literature. We added corresponding references for the layer selection in the method section (line 687).

In summary, the presentation of the propose generative framework seems a bit premature with many open questions not being addressed. This includes a demonstration of the framework's scientific value.

We thank the reviewer for the constructive criticism. We extensively revised the manuscript, validated all key results on a much larger video sample, added comparisons to previous algorithms, added real-world applications of the proposed algorithm with three psychophysical experiments in humans, and introduced a new stimulus generation procedure based on the proposed algorithm, that for the first time allows to spatiotemporally mix two videos. We hope that the reviewer agrees that the revised manuscript now well demonstrates the framework's scientific value.

References:

Portilla/Simoncelli (2000), A Parametric Texture Model Based on Joint Statistics of Complex Wavelet Coefficients, *International Journal of Computer Vision* 40(1), 49–71

Reviewer #3

The paper presents a Spatiotemporal Style Transfer (STST) algorithm for generating dynamic visual stimuli in vision research. The STST algorithm is based on a two-stream deep neural network model that separates spatial and temporal features to synthesize video stimuli. This approach allows researchers to create "model metamers," dynamic stimuli that match the low-level spatiotemporal features of natural videos but lack high-level semantic content. The study demonstrates that the generated stimuli can be used to investigate object recognition and probe internal representations in predictive coding networks.

Novelty

The STST algorithm represents a significant advancement in the field of dynamic visual stimulus generation, filling a gap left by previous methods that primarily focused on static images. By integrating both spatial and temporal aspects, this work offers a novel approach to potentially further understanding visual information processing in both artificial and biological systems.

We thank the reviewer for the positive assessment of our work.

However, the algorithm employed to generate the stimuli itself seems to be a straightforward extension of neural style transfer to videos, lacking sufficient novelty in methodology and appropriate discussion of prior work. Meanwhile, although the claimed objective of the paper appears novel, it does not sufficiently demonstrate an application of the proposed approach in directly studying biological systems, as all experiments were validated by neural networks rather than through subjective studies and psychophysical evaluations.

We thank the reviewer for highlighting these important points. Following the reviewer's suggestion, we performed three new psychophysical studies in humans using stimuli generated with the STST algorithm. The first and second experiments investigate the effect of STST metamer stimuli on object recognition and compare STST and STPS algorithms for low-level features preservation in humans, respectively. The third experiment introduces an entirely new and unique use case of the STST algorithm, which is to generate stimuli that factorize the spatial and temporal features from two different target videos, i.e. that mix the spatial and temporal features from two different movies. To the best of our knowledge, the generation of such stimuli was previously impossible. We provide new results on both humans and deep vision models on these stimuli, that suggest a spatial bias in visual similarity judgements in both artificial and biological systems. We included all these new experiments and results in the revised manuscript. Furthermore, we followed the reviewer's suggestion and added a new section in the discussion, in which we elaborate on the video style transfer literature and its relation to the proposed algorithm (line 569).

Methodology, Clarity, and Presentation

The paper is well-organized, with a logical flow from introduction to methodology, results, and discussion. The figures effectively illustrate the STST algorithm and its outputs.

In Eqn 4, the notations L_{sc} and L_{st} were never introduced. I assume they refer to “spatial content” and “spatial texture” respectively?

We thank the reviewer for this comment. We provide definitions for these terms starting from line 164: “[...] Moreover, α and β represent the weights associated with the content \mathcal{L}_{sc}^{ℓ} and texture \mathcal{L}_{st}^{ℓ} loss in the spatial loss $\mathcal{L}_{spatial}$, as well as θ and λ are the weights for the content \mathcal{L}_{tc}^{ℓ} and texture \mathcal{L}_{tt}^{ℓ} loss in the temporal loss $\mathcal{L}_{temporal}$.”

The total variation loss introduced by Eqn 9 does not seem to preserve edges, as large variation over edges can also be penalized by Eqn 9.

We thank the reviewer for bringing this up. We agree that the total variation loss counteracts edges. This is why we opted for a moderate effect of this loss component by using a comparatively low hyperparameter weight. Furthermore, we performed and added an ablation study on the effect of the total variation loss in the revised manuscript (Fig. S2, from line 292).

The result images presented in the paper are too small to understand the content effectively. The supplementary materials also did not include any videos processed by the authors for evaluation. Only the source video of the animal scene has been included.

We thank the reviewer for pointing this out. We apologize for the inconvenient size of the images, which however followed the Nature guidelines on figure formatting. We added all original and generated videos to the supplementary materials.

Evaluation

The number of videos used for evaluation and validation is too small and subject to cherry-picking. The authors are advised to select a public dataset without any biased manual selection of content for a fair and extensive evaluation.

We thank the reviewer for highlighting this important point. We followed the reviewer’s suggestion, and validated all our key results on the spatiotemporal feature analysis, deep vision model representations and PredNet performance on a larger validation dataset comprising 100 videos randomly sampled from the Kinetics400 dataset (see results section). All our previous results were confirmed. Furthermore, we extended our analyses by adding a comparison with STPS videos on the deep vision model representations and PredNet performance.

Meanwhile, the hypotheses proposed by the authors have only been validated by neural networks. Since the paper aims to apply the proposed generated stimuli to test specific hypotheses in both biological and artificial systems, I was wondering if there are any hypotheses that can be validated through rigorous psychophysical experiments.

We thank the reviewer for bringing this up. Following the reviewer’s suggestion, we performed three new psychophysical experiments in humans. In the first and second experiment we show that also in biological systems the high-level content of the metamer stimuli for object recognition was not recognized and that the STST algorithm better preserved the low-level features than the comparison algorithm. Furthermore, with the third experiment we introduce a new spatiotemporal factorization procedure based on the STST algorithm (from line 470) to study the interaction between spatial and temporal features in both human and artificial visual systems. Our results suggest that human similarity judgements and representational similarities in deep vision models are biased towards spatial as compared to temporal

features, providing insights into the representational capabilities of biological and artificial systems for the spatiotemporal integration of dynamic visual information.

Finally, an ablation study for the total variation loss and the color matching method is highly recommended but currently missing.

We thank the reviewer for highlighting this important point. We followed the reviewer's suggestion and performed an ablation study to investigate the effect of the color matching step and of the total variation loss. The results show that color matching has a strong impact on contrast and luminosity, as well as on the full color distribution, while absence of the total variation loss induces excessive high-frequency noise (Fig. S2 and from line 292).

Remarks on code availability:

I ran into the following error when attempting to run "python synthesize.py -gpu 1 -tar Animal.mp4"

Traceback (most recent call last):

File "STST/synthesize.py", line 115, in <module>

gen = stst.generate(config)

^^^^^^^^^^^^^^^^^^^^^^^^^^^^^^^^

File "STST/SpaceTimeStyleTransfer.py", line 384, in generate

log = open(self.logname,'w')

^^^^^^^^^^^^^^^^^^^^^^^^^^^^^^^^

FileNotFoundError: [Errno 2] No such file or directory: 'log/log_240607_1856_.txt'

We thank the Reviewer for highlighting this. We checked and found that the only problem was a missing a folder (the "log" folder) in the environment, which prevented the creation of the log file. We updated the supplementary software and successfully tested it.

We thank the reviewers and the editor for their positive assessment of our work, and for their thoughtful and constructive comments. These comments were very helpful for improving the manuscript. Below, we provide point-by-point responses to all three reviewers. To ease navigation, reviewer comments are written in **black** and our replies in **blue**.

Reviewer #1 (Remarks to the Author):

I appreciate the additional analyses and experiments in this revision, which has addressed my previous comments. Furthermore, the psychophysics results are instructive, and the experiment on factorizing spatiotemporal information is very nice. Below are some comments on the new material.

We thank the reviewer for the positive assessment of our work.

1. While I appreciate the expanded analyses in Fig. 4, I do not follow the reasoning surrounding SSIM, PredNet, and high-level information. The SSIM metric is not expected to capture high-level information, so how do the SSIM results show that high-level information plays no role in PredNet? The results could be because the low-level information in STST and STPS videos is easier (per SSIM) for PredNet to capture than the low-level information in natural videos. This does not show whether or not PredNet captures high-level information.

We thank the reviewer for pointing this out. We agree that SSIM does not capture high-level content in images, but our reasoning is not based on this. We used the SSIM metric to quantify the quality of model predictions about the next frames. The reason why we claim our results are related to the lack of high-level content is because the video inputs were manipulated as such. If PredNet exploited a high-level semantic understanding of videos for its next frame prediction, it should make less errors in predicting meaningful natural videos as compared to metamer videos that lack high-level content. Crucially, we found exactly the opposite, which is that PredNet was even significantly worse at natural videos compared to STST and STPS videos. This means that, for example in the video with the turtle swimming, PredNet is not able to better predict the motion of the object than the same motion not linked to an object in metamer videos. We believe this is critical evidence in favor of the hypothesis that PredNet acts more as a flow filter rather than predicting the next frame at higher levels.

Furthermore, we agree with the reviewer's intuition that low-level features could be more easily predicted in STST and STPS videos as compared to natural ones. This is indeed what our results show (Fig. 3d). However, this effect is not because of the low-level feature content *per se*, which, as we demonstrate, is very similar for natural and metamer stimuli. Instead our results indicate that this effect is driven by the higher frame-to-frame similarity for metamer videos compared to natural ones (Fig. 3e and 5f). But again, this is no evidence that PredNet is using high-level content to predict.

We carefully revised the corresponding text in the results and discussion sections to clarify our reasoning.

2. I'm unsure that the results in Fig. 7 show people are biased for spatial features in similarity judgments. The synthetic videos based on current methods may be better at mimicking spatial features than temporal features. This possibility could also explain the analogous results in

models. Moreover, in the results for models, the absolute accuracy difference is small on the left side of Fig. 7e and f (< 0.06 and < 0.01 respectively).

We thank the reviewer for highlighting this point. We agree that we cannot entirely rule out the possibility that a relative discrepancy between metamer and target videos for spatial and temporal features may affect our results. This discrepancy can be noted in figure 5e. We also added the average correlation of the factorized metamers in space and time for both stimulus sets, which now reads:

“For both stimulus sets, the spatial and temporal features of the 6 mixed metamer movies were correlated with either the spatial (average correlation in stimulus sets 1/2, $r = 0.98/0.95$) or temporal (average correlation in stimulus sets 1/2, $r = 0.90/0.78$) natural target movie, respectively (Fig. 5e).”

Nevertheless, a critical finding suggests that this discrepancy cannot explain the reported spatial bias. For both humans and deep vision models, accuracy and CKA scores in the temporal task are higher for stimulus set 2 compared to stimulus set 1. However, the average correlation of temporal features with their targets is higher in stimulus set 1 (0.90) than in stimulus set 2 (0.78). This is opposite what would be expected if a lower correlation of temporal features was driving the lower temporal accuracy and CKA scores. We included a discussion of this point in the discussion section, which now reads:

“A possible confound of these findings is the relative discrepancy between metamer and target videos for spatial and temporal features. Specifically, in the factorized metamers spatial features were slightly better matched with their natural targets than temporal features. Moreover, the average correlation of temporal features with their targets was higher in stimulus set 1 (0.90) than in stimulus set 2 (0.78). Nevertheless, in the temporal task, both humans and deep vision model had higher accuracy and CKA scores for stimulus set 2 than for set 1. This provides evidence against the hypothesis that the discrepancy between metamer and target features explains the results.”

Minor comments:

1. Regarding whether the normalization factor of $1/N_l$ is extraneous in Eqn. (2), substituting Eqn. (2) into Eqn. (3) suggests that the sum is normalized by a total factor of N_l^4 , whereas the sum is only over N_l^2 elements. It would be helpful for the authors to share an intuition as to why.

We thank the reviewer for pointing this out. N_ℓ refers to the number of filters in a convolutional layer. Eq. 2 represents the computation of the Gram matrix G^ℓ , which requires to be normalized by both the number of filters N_ℓ and the number of features in the feature map M_ℓ . In eq. 3, the texture loss is basically a mean squared error loss between the Gram matrix of the natural video and the generated one. Thus, since the Gram matrix has dimensionality $N_\ell \times N_\ell$, the normalization factor is $N_\ell \times N_\ell$. The factor 1/2 is added by convention to take into account the derivative applied during optimization.

2. Line 220: Should the H_l and W_l be indexed by o (calligraphic o) instead of σ (sigma)? It looks like the calligraphic o is intended to be the index for the octave levels.

We thank the reviewer for pointing this out. Indeed, this was a mistake in the indexing that we corrected now.

Reviewer #2 (Remarks to the Author):

In my assessment of the original submission, my main concern was the lack of a thorough and convincing demonstration of the value and usefulness of the proposed generative method. The authors have addressed this with the inclusion of new experimental data and a revised and extended discussion. The work has substantially improved; I have no further comments.

We thank the reviewer for the positive assessment of our work.

Reviewer #3 (Remarks to the Author):

The revised manuscript has effectively addressed most of my concerns, improving both the depth of the analysis and the clarity of the presentation. The three newly added psychophysical experiments in humans provide evidence supporting the effectiveness of the proposed Spatiotemporal Style Transfer (STST) algorithm for studying biological systems.

We thank the reviewer for the positive assessment of our work.

A minor concern is that the conclusion drawn from the third experiment regarding the spatial bias appears to rely on the assumption that the metamers generated by STST are the ground truth. However, previous experiments only demonstrate that STST performs better in preserving low-level features than existing methods, rather than proving its results are exact and error-free. I hope other reviewers with expertise in psychophysics can further comment on this.

We thank the reviewer for highlighting this point. We agree that we cannot entirely rule out the possibility that a relative discrepancy between metamer and target videos for spatial and temporal features may affect our results. This discrepancy can be noted in figure 5e. We also added the average correlation of the factorized metamers in space and time for both stimulus sets, which now reads:

“For both stimulus sets, the spatial and temporal features of the 6 mixed metamer movies were correlated with either the spatial (average correlation in stimulus sets 1/2, $r = 0.98/0.95$) or temporal (average correlation in stimulus sets 1/2, $r = 0.90/0.78$) natural target movie, respectively (Fig. 5e).”

Nevertheless, a critical finding suggests that this discrepancy cannot explain the reported spatial bias. For both humans and deep vision models, accuracy and CKA scores in the temporal task are higher for stimulus set 2 compared to stimulus set 1. However, the average correlation of temporal features with their targets is higher in stimulus set 1 (0.90) than in stimulus set 2 (0.78). This is opposite what would be expected if a lower correlation of temporal features was driving the lower temporal accuracy and CKA scores. We included a discussion of this point in the discussion section, which now reads:

“A possible confound of these findings is the relative discrepancy between metamer and target videos for spatial and temporal features. Specifically, in the factorized metamers spatial features were slightly better matched with their natural targets than temporal features.

Moreover, the average correlation of temporal features with their targets was higher in stimulus set 1 (0.90) than in stimulus set 2 (0.78). Nevertheless, in the temporal task, both humans and deep vision model had higher accuracy and CKA scores for stimulus set 2 than for set 1. This provides evidence against the hypothesis that the discrepancy between metamer and target features explains the results.”

In addition, the evaluation on a larger and more diverse dataset, along with the addition of the ablation study, is also appreciated. Overall, the revisions have significantly enhanced the manuscript, and I recommend acceptance after addressing these remaining points.

We thank the reviewer for the positive assessment of our work.